# The neural basis for a persistent internal state in *Drosophila* females

**David Deutsch[1], Diego Pacheco[1], Lucas Encarnacion-Rivera[1], Talmo Pereira[1], Ramie Fathy[1], Jan Clemens[1†], Cyrille Girardin[1‡], Adam Calhoun[1], Elise Ireland[1], Austin Burke[1], Sven Dorkenwald[1,2], Claire McKellar[1], Thomas Macrina[1,2], Ran Lu[1], Kisuk Lee[1,3], Nico Kemnitz[1], Dodam Ih[1], Manuel Castro[1], Akhilesh Halageri[1], Chris Jordan[1], William Silversmith[1], Jingpeng Wu[1], H Sebastian Seung[1,2], Mala Murthy[1]\***

[1]Princeton Neuroscience Institute, Princeton University, Princeton, United States; [2]Department of Computer Science, Princeton University, Princeton, United States; [3]Brain & Cognitive Science Department, Massachusetts Institute of Technology, Cambridge, United States

**\*For correspondence:**
mmurthy@princeton.edu

**Present address:** †European Neuroscience Institute, Max Planck, Göttingen, Germany; ‡State Secretariat for Education, Research and Innovation SERI, Switzerland

**Competing interests:** The authors declare that no competing interests exist.

**Abstract** Sustained changes in mood or action require persistent changes in neural activity, but it has been difficult to identify the neural circuit mechanisms that underlie persistent activity and contribute to long-lasting changes in behavior. Here, we show that a subset of Doublesex+ pC1 neurons in the *Drosophila* female brain, called pC1d/e, can drive minutes-long changes in female behavior in the presence of males. Using automated reconstruction of a volume electron microscopic (EM) image of the female brain, we map all inputs and outputs to both pC1d and pC1e. This reveals strong recurrent connectivity between, in particular, pC1d/e neurons and a specific subset of Fruitless+ neurons called aIPg. We additionally find that pC1d/e activation drives long-lasting persistent neural activity in brain areas and cells overlapping with the pC1d/e neural network, including both Doublesex+ and Fruitless+ neurons. Our work thus links minutes-long persistent changes in behavior with persistent neural activity and recurrent circuit architecture in the female brain.

## Introduction

Social behaviors are known to be affected by persistent internal states (*Anderson, 2016*; *Berridge, 2004*; *Lorenz and Leyhausen, 1973*). These states correspond with levels of arousal or drive, and can impact whether and how individuals interact, with consequences for mating decisions and reproduction (*Chen and Hong, 2018*; *Kennedy et al., 2014*; *Stowers and Liberles, 2016*). The neural mechanisms underlying arousal states remain largely unknown.

In male flies, a small population of male-specific neurons (P1) that express the sex-specific transcription factors Fruitless and Doublesex (*Auer and Benton, 2016*), drive both male-aggression and male-mating behaviors (*Hoopfer et al., 2015*; *Koganezawa et al., 2016*; *von Philipsborn et al., 2011*). Brief optogenetic activation of P1 neurons drives both persistent song production in solitary males and persistent aggression upon introduction of another male, both over minutes (*Bath et al., 2014*; *Hoopfer et al., 2015*; *Inagaki et al., 2014*). While P1 activation is sufficient for eliciting the persistent behavioral phenotypes, other groups of neurons, such as pCd (also called pC3 [*Rideout et al., 2010*]), are involved in maintaining the persistent state (*Jung et al., 2020*; *Zhang et al., 2019*) - these neurons are not synaptically coupled to P1, and the circuit architecture that mediates persistence remains unresolved. P1 neurons also receive dopaminergic input that affects mating drive over longer timescales of hours (*Zhang et al., 2016*), suggesting neuromodulation is also important for persistent changes in behavior driven by P1. Work on P1 in flies bears

**eLife digest** Long-term mental states such as arousal and mood variations rely on persistent changes in the activity of certain neural circuits which have been difficult to identify. For instance, in male fruit flies, the activation of a particular circuit containing 'P1 neurons' can escalate aggressive and mating behaviors. However, less is known about the neural networks that underlie arousal in female flies. A group of female-specific, 'pC1 neurons' similar to P1 neurons could play this role, but it was unclear whether it could drive lasting changes in female fly behavior.

To investigate this question, Deutsch et al. stimulated or shut down pC1 circuits in female flies, and then recorded the insects' interactions with male flies. Stimulation was accomplished using optogenetics, a technique which allows researchers to precisely control the activity of specially modified light-sensitive neurons.

Silencing pC1 neurons in female flies diminished their interest in male partners and their suitor's courtship songs. Activating these neural circuits made the females more receptive to males; it also triggered long-lasting aggressive behaviors not typically observed in virgin females, such as shoving and chasing.

Deutsch et al. then identified the brain cells that pC1 neurons connect to, discovering that these neurons are part of an interconnected circuit also formed of aIPg neurons – a population of fly brain cells that shows sex differences and is linked to female aggression. The brains of females were then imaged as pC1 neurons were switched on, revealing a persistent activity which outlasted the activation in circuits containing both pC1 and aIPg neurons. Thus, these results link neural circuit architecture to long lasting changes in neural activity, and ultimately, in behavior. Future experiments can build on these results to determine how this circuit is activated during natural social interactions.

some similarity to work in mice. In male mice, optogenetic activation of SF1 (steroidogenic factor 1) expressing neurons in the dorsomedial part of the ventromedial hypothalamus (VMHdm$^{SF1}$) drive multiple defensive behaviors that outlast the stimulation by up to 1 min (*Kunwar et al., 2015*; *Wang et al., 2015*). In addition, activity of VMHdm$^{SF1}$ neurons can persist for over a minute following the presentation of a fear stimulus. Computational modeling work suggests a mix of recurrent circuitry and neuromodulation underlie the persistence (*Kennedy et al., 2020*), but this has not yet been tested.

We know little about the persistence of social behaviors in females across taxa. P1 neurons are a subset of the larger Doublesex+ pC1 neural subset (*Kimura et al., 2008*). While *Drosophila* females lack P1 neurons, they have Doublesex+ pC1 neurons (*Rideout et al., 2010*; *Robinett et al., 2010*; *Zhou et al., 2014*), including a subset that are female-specific (*Wu et al., 2019*). Activation of pC1 neurons affects receptivity toward males (*Wang et al., 2020b*; *Zhou et al., 2014*), drives chasing of males (*Rezával et al., 2016*; *Wu et al., 2019*), and aggressive behaviors toward females (*Palavicino-Maggio et al., 2019*; *Schretter et al., 2020*). These data suggest that female pC1 neurons can drive an arousal state, similar to male P1 neurons, but whether female pC1 neurons can drive *persistent* changes in behavior and *persistent* neural activity, has not yet been investigated. Importantly, a complete electron microscopic volume is currently available for the entire adult female brain (*Zheng et al., 2018*), making it possible to link brain activity to complete wiring diagrams in females. Direct measurements of synaptic connectivity can determine whether recurrent neural networks, known to be important for persistent neural activity lasting for seconds (*Aksay et al., 2007*), also underlie minutes-long persistent activity and changes in behavior, as has been recently proposed for male mice and flies (*Jung et al., 2020*; *Kennedy et al., 2020*).

Here, we show that pC1 activation drives persistent changes in female behavior for minutes following stimulus offset, and we identify the subset of pC1 neurons (called pC1d/e [*Wang et al., 2020a*]; also referred to as pC1-Alpha [*Wu et al., 2019*]) that affects the persistent aggressive and male-like behaviors. A companion study *Schretter et al., 2020* demonstrates that pC1d, but not pC1e, drives female aggressive behaviors. By leveraging the automated segmentation of an electron microscopic volume of the female brain (*Dorkenwald et al., 2020*; *Zheng et al., 2018*), we map all inputs and outputs of both pC1d and pC1e and find a strong recurrent neural network with

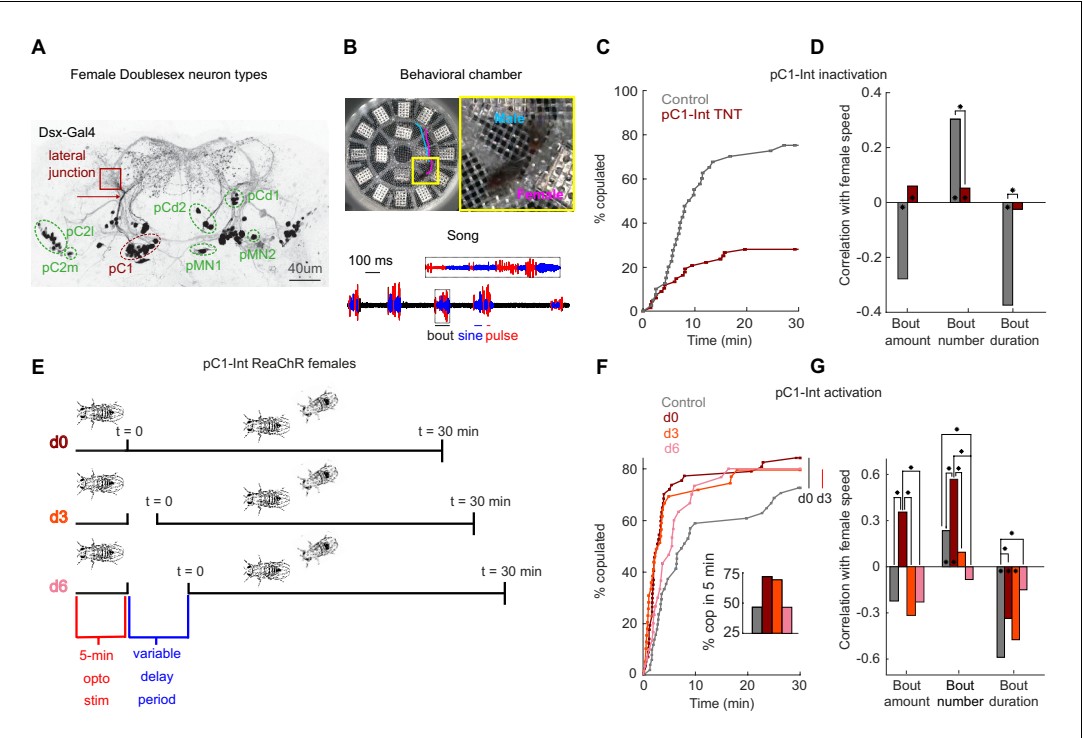

**Figure 1.** pC1-Int activation has a persistent effect on female receptivity and responses to male courtship song. (A) Dsx+ neurons in the female central brain. Max z-projection of a confocal stack in which Dsx+ cells are labeled with GFP (adapted from *Deutsch et al., 2019*). Dsx is expressed in eight morphologically distinct cell types in the female brain (seven types are indicated by circling of their somas; the more anterior cell type aDN [*Lee et al., 2002*] is not shown). pCd has two morphologically distinct types, pCd1 and pCd2 (*Kimura et al., 2015*). Many of these cells project to a brain region known as the lateral junction (red square). pC1 cells project to the lateral junction through a thin bundle (red arrow). This bundle was used to identify these cells in the EM dataset (see *Figure 3—figure supplement 1*). (B) Behavioral chamber (diameter ~25 mm) tiled with 16 microphones to record song. Male and female positions were tracked (a 1.5 s example trace is shown for the male (cyan) and female (magenta)), and male song was automatically segmented into sine and pulse (below). (C) Percent of male/female pairs that copulated as a function of time. Cox proportional hazards regression p=6.3*10$^{-6}$: pC1-Int TNT (red, n = 68 pairs) compared with controls (gray, n = 40 pairs). (D) Rank correlation between male song (bout amount, number and duration) and female speed for pC1-TNT females (dark red) or control females (gray) paired with wild-type males. Significance between experimental groups was measured using ANOCOVA and multiple comparison correction (*p<0.01). An asterisk on a bar indicates a significant correlation between a single male song measure and female speed (*p<0.01). (E) Experimental design for pC1-Int activation. pC1-Int cells were activated (using ReaChR) for 5 min in a solitary female placed in the behavioral chamber. Following light offset, a wild-type male was introduced at t = 0, following a variable delay period (d0 = no delay; d3 = 3 min delay; d6 = 6 min delay). All behavioral phenotypes were measured at t > 0. (F) Same as (C), but for females expressing ReachR in pC1-Int cells according to the protocol shown in (E). Inset: The percent of pairs copulated between t = 0 and t = 5 min for each condition. pC1-Int activated females in the d0 condition (n = 57) copulated significantly faster than controls (n = 51; vertical black line; p=0.0045, Cox's proportional hazards regression model, accounting for censoring, as not all flies copulated in 30 min; black vertical line). Time to copulation was also shorter in the d3 group (n = 39 pairs) compared with controls (d0, no ATR), but the difference was not significant after Bonferroni correction (p=0.034; red vertical line), and no significant difference was found between the 6 min delay (n = 30) and control groups (p=0.21). (G) Same as (D), but for females expressing ReachR in pC1-Int cells according to the protocol shown in (E). Asterisks show significance, using the same criteria as in (D). Numbers of pairs are the same as in (F).

The online version of this article includes the following figure supplement(s) for figure 1:

**Figure supplement 1.** Tracking behavior with LEAP, receptivity of pCd1 inactivated and activated females, and song production of males paired with pC1-Int activated females.

Fruitless+ aIPg neurons. Using pan-neuronal calcium imaging, we find that pC1d/e activation can elicit persistent activity for minutes among multiple cell types. The persistent activity is present in Doublesex+ and Fruitless+ neurons, including pC1 neurons themselves. We thus link minutes-long persistent neural activity and behavior with reciprocal connectivity in the female brain.

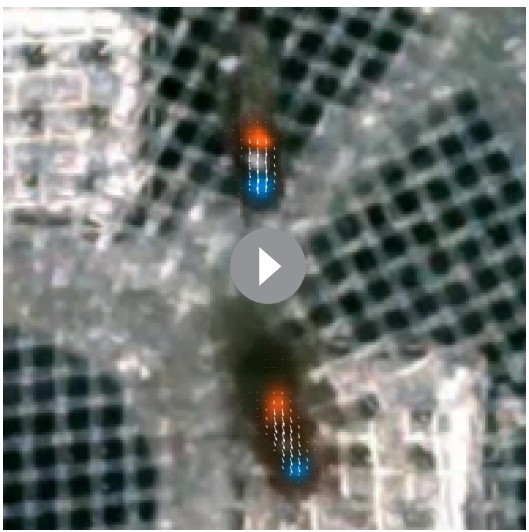

**Video 1.** Confidence maps (head in blue, thorax in orange) and part affinity vector fields (white arrows) calculated using LEAP (*Pereira et al., 2019*) for the male and female. The male has a white painted dot on his back. Male chasing and singing are shown, as well as female shoving. Movie is slowed down four times.
https://elifesciences.org/articles/59502#video1

## Results

### Female pC1 activation persistently modulates both female receptivity and song responses

To investigate the neural basis of a persistent internal state in the female brain, we focused on pC1 neurons, one of eight Doublesex-expressing cell types in the central brain (*Figure 1A*; *Kimura et al., 2015*). We used an intersection between two driver lines (Dsx-GAL4 and R71G01-LexA, hereafter referred to as pC1-Int; see Fly genotype table for list of genotypes used in this study), to label pC1 neurons, as done previously (*Rezával et al., 2016*; *Zhou et al., 2014*) – prior work shows this line labels, per hemisphere, ~6 neurons, all of which project to the lateral junction (*Zhou et al., 2014*). We tracked male and female body parts (head and thorax) in addition to recording all sounds (*Figure 1B*, *Figure 1—figure supplement 1A*, *Video 1*; see Materials and methods for details on song segmentation and tracking of flies on a non-homogenous background). Silencing pC1-Int neurons in females affects receptivity *Zhou et al., 2014*; we corroborated these results (*Figure 1C*) and additionally showed that silencing pC1-Int neurons

diminished responses to male song (*Figure 1D*). Persistent changes in the behavioral state of *Drosophila* males have been studied by examination of social behaviors following optogenetic activation of P1 neurons (*Hoopfer et al., 2015*; *Jung et al., 2020*). We activated pC1-Int in a solitary virgin female for 5 min, followed by a variable delay period, after which a virgin male was introduced to examine female behaviors in the context of courtship (*Figure 1E*) - there was no optogenetic activation following the first 5 min. The activity of stimulated neurons should decay during the variable delay period (d0 (0 min delay), d3 (3 min delay), or d6 (6 min delay)) – below, we test this explicitly with neural imaging, and also examine shorter activation periods. This paradigm therefore allowed us to examine the effects of differing levels of persistent activity on behavior.

Experimental flies were fed all-trans-retinal (ATR), which is required for ReaChR (red-shifted Channelrhodopsin) function in flies (*Inagaki et al., 2014*). Control flies shared the same genotype but were not fed ATR. We found that activation of pC1-Int neurons induces a persistent effect on female receptivity and responses to male song, but that this effect diminishes with a delay period between neural activation and introduction of a male. pC1-Int-activated females copulated significantly faster than controls in the d0 condition, with reduced copulations following a delay between optogenetic activation and introduction of a male (*Figure 1F*). About 75% of the flies copulated within 5 min in the d0 and d3 conditions, compared with fewer than 50% in the control and d6 groups (*Figure 1F*, inset). Another set of Dsx+ neurons called pCd1 (*Figure 1A*) was also shown to control female receptivity (*Zhou et al., 2014*) and, additionally, to enable P1-induced persistent activity in males (*Jung et al., 2020*). While we confirmed that pCd1 silencing in females reduced receptivity (*Figure 1—figure supplement 1B*), we found that 5 min pCd1 activation had no persistent effect on female receptivity (*Figure 1—figure supplement 1C*).

Activating pC1-Int also produced a persistent effect on responses to male song, overall with the strongest effect at the d0 delay (*Figure 1G*) - d0 females, in comparison with controls, accelerated more in response to all song elements, behaving like unreceptive females (*Coen et al., 2014*). This effect was not due to changes in wild-type male song structure for the d0 condition, although male song bouts were shorter and less frequent in the d3 and d6 conditions (*Figure 1—figure supplement 1D*), possibly due to the strong effect of pC1-Int activation on male-female interactions at longer delays, as shown below. Because at d0, females are both hyper-receptive (mate quickly) but also

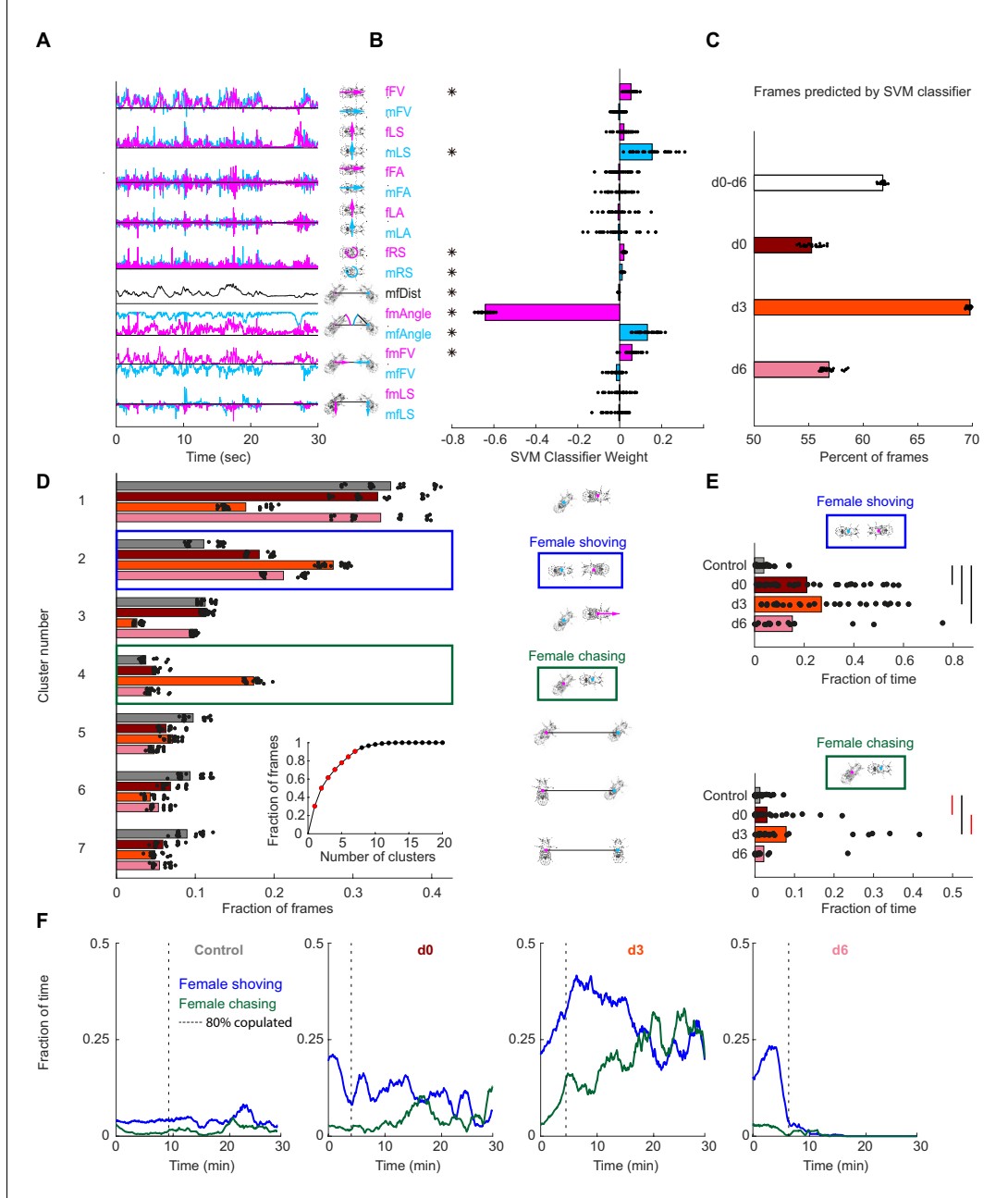

**Figure 2.** Automated identification of persistent female behaviors following pC1-Int activation. (**A**) For each video frame, 17 parameters were extracted based on tracking of male/female position and heading (see Materials and methods for parameter definition). An example trace (30 s) is shown for each parameter. (**B**) 30 independent (using non-overlapping sets of video frames) Support Vector Machine (SVM) classifiers were trained to classify frames (each frame represented by 17 values) as belonging to control or experimental group (all delays d0-d6 considered together; see Materials and methods). Each classifier is represented by 17 points - one for each parameter. Each point is the weight associated with a given parameter for one classifier, and the bar height represents the mean over classifiers (*p<$10^{-4}$, one-sample t-test with Bonferonni correction for multiple comparisons; see Materials and methods). (**C**) The percent of frames correctly classified using the SVM classifier. Each dot is the prediction of a single SVM classifier, trained to classify frames as belonging to control or experimental group (d0, d3, d6, or d0-d6 together) – 30 classifiers and their mean plotted for each group. (**D**) K-means was used to cluster frames based on the eight most significant parameters (marked with asterisks in (**B**)). The largest 7 clusters include 90.4% of the frames (see inset). Clustering was performed 30 times (black dots; bars = mean), using different but overlapping sets of frames. The same number of frames was taken from each group (see Materials and methods). Cluster 2 (blue box - 'female shoving') is more probable following pC1-Int activation (in d0, d3 and d6 conditions) compared to control, while cluster 4 (green box- 'female chasing') is more probable in the d3 condition only compared to control. At right, schematic describing the male-female interaction in each cluster, based on the mean values of the weights. (**E**) JAABA-based classification of shoving (top) and chasing (bottom) behaviors. Each dot represents a single pair of flies. The fraction of time the male-

*Figure 2 continued on next page*

*Figure 2 continued*

female pair spent shoving (0.037/0.21/0.27/0.15 for control/d0/d3/d6) or chasing (0.013/0.030/0.079/0.022) are shown. Black lines represent significant differences with p<0.05 after Bonferroni correction for multiple comparison. Red lines - significant before, but not after correction for multiple comparisons. (F) Fraction of time females spent chasing or shoving (moving average with a 2-min window), based on JAABA classification in each condition (control, d0, d3, d6). T = 0 is the time the male was introduced (see *Figure 1E*), and the vertical dashed line indicates the time, for each condition, when 80% of the pairs copulated. Behaviors are not scored after copulation.

The online version of this article includes the following figure supplement(s) for figure 2:

**Figure supplement 1.** Detection of female shoving and chasing from male and female movements.
**Figure supplement 2.** Female shoving and chasing detected using machine learning, and validated by manual inspection.
**Figure supplement 3.** Manually-detected persistent behaviors in females, following pC1-Int activation.

accelerate in response to song, we suspect that different subsets of pC1 neurons may control these two effects.

## Female pC1 activation drives persistent female shoving and chasing

To quantify other behaviors elicited by pC1-Int activation, we used an unsupervised approach, and first decomposed male and female movements and interactions into 17 parameters (*Calhoun et al., 2019*; *Figure 2A*). We then used a Support-Vector Machine (SVM) framework (*Cortes and Vapnik, 1995*; *Cristianini and Shawe-Taylor, 2000*) to find the weights that best classify single frames as belonging to sessions of control vs experimental groups (all delay conditions, d0-d6). We found that the weights of 8 out of 17 parameters were significantly different from zero (*Figure 2B* and *Figure 2—figure supplement 1A*), with the strongest weight being fmAngle, defined as the degrees the female needs to turn in order to point toward the male centroid. The weight of fmAngle is negative because this parameter is smaller in the experimental flies compared with controls, indicating that pC1-Int activated females spend more time facing the male (*Figure 2—figure supplement 1B*). When separated by experimental condition, the SVM classifier performed best on the d3 condition versus control (*Figure 2C*), indicating that male-female movements and interactions are most distinct following this delay.

Next, we clustered individual video frames based on the values of the eight parameters identified as most important by the SVM (*Figure 2—figure supplement 1A*, asterisks), and found that the largest seven clusters accounted for over 90% of all frames (*Figure 2D*, inset). The weights of the eight parameters were different for each cluster (*Figure 2—figure supplement 1C*), representing different behaviors. Five of the clusters describe behaviors that are the same or reduced following pC1-Int activation (clusters 1 and 3, male chasing female, and 5–7, increased male-female distance). Two clusters, however, describe behaviors that occur with higher probability following pC1-Int activation (clusters 2 and 4; *Figure 2D*). Cluster 2 is characterized by small fmAngle and small mfAngle (indicating that the male and female are facing each other), decreased male-female distance, and large fmFV (indicating that the female is close to the male and moving in his direction). Cluster 4 is characterized by small fmAngle and large mfAngle (indicating that the female is behind the male), decreased male-female distance, and large fmFV (indicating that the female is moving in the direction of the male). Based on the weight values, and verified by inspection of the videos following

**Video 2.** A sequence of female shoving and chasing. The female shoves the male while occasionally extending one or two wings, and then chases the male while occasionally extending a single wing or contacting the male with her front legs. Finally, the male attempts to copulate, the female spreads her wings, and copulation occurs. Movie is in real time. Experimental condition: pC1-Int, d0.
https://elifesciences.org/articles/59502#video2

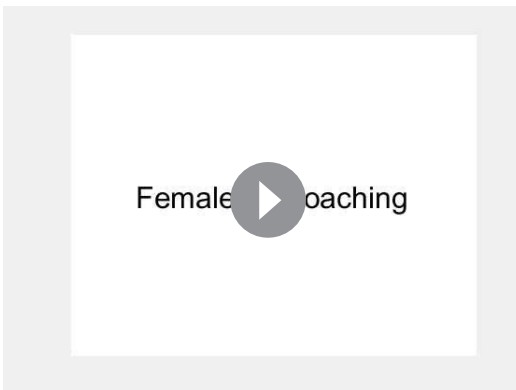

**Video 3.** Multiple example behaviors: female approaching (*Figure 2—figure supplement 3A*), shoving and circling (*Figure 2—figure supplement 3B*), female headbutting and female mounting (*Figure 2—figure supplement 3C*). Following female approaching in this example, there is a short epoch of circling. In the 'shoving and circling' example, the female shoves the male before a circling behavior starts (See *Figure 2—figure supplement 3B*, inset). In the female headbutting example, the female extends two wings while headbutting the male, followed by a male jump. In the 'female mounting' example, the female positions herself behind the male and climbs on his back. Circling and headbutting examples are from pC1-Int (d0) condition, and female approaching/mounting from pC1-Int (d3) condition.
https://elifesciences.org/articles/59502#video3

clustering (*Video 2*), we termed cluster 2 'female shoving' and cluster 4 'female chasing'. For both female shoving and female chasing, the amount of each behavior was highest in the d3 condition relative to control (*Figure 2D*).

We then used JAABA (*Kabra et al., 2013*) to train a classifier to recognize epochs (groups of video frames) of female chasing and female shoving in the data (*Figure 2—figure supplement 1D*; see Materials and methods). This analysis confirmed that the amount of female shoving and chasing was greatest in the d3 condition relative to control (*Figure 2E*), in addition to revealing that the duration of female chasing and shoving bouts was longer (*Figure 2—figure supplement 1E*). In many cases females transitioned directly between shoving and chasing, and the conditioned probability (transition probability, given that a transition occurred) for shoving →chasing was more than double at the d3 condition compared to the d0 and d6 conditions. By examining shoving and chasing probabilities over time (*Figure 2F*), following the introduction of a male (t = 0), we found that both behaviors persisted for as long as 30 min in the d3 condition (*Figure 2F*), but not in the d0 and d6 conditions, suggesting a complex interaction between neural activity in the female brain at the time of introduction of the male and feedback or social cues. While the percent of time females spent shoving or chasing in the first two minutes after the male was introduced was similar in the d0 and d3 conditions (19.7/21.3% for shoving, 2.8/3.1% for chasing), shoving and chasing probabilities rose over time in the d3, but not in the d0 condition. In the d6 condition, shoving probability was comparable to the probability in the d0-d3 conditions in the first two minutes (15%), but decayed to control level after 6 min. The transition into 'chasing' from 'shoving' or from 'other' (no shoving and no chasing) epochs also peaked at d3 (*Figure 2—figure supplement 2*). In all cases, we only examined female behaviors prior to copulation. We also activated another Dsx+ cell type, pCd1, which is part of a circuit that drives persistent behavior in the male brain (*Jung et al., 2020*), but observed neither shoving nor chasing following 5 min of pCd1 activation in females (*Figure 2—figure supplement 2A*). This suggests that the persistent behavioral effects we observe are specific to pC1 activation.

Manual inspection of the videos identified several additional behaviors produced by females following pC1-Int activation (*Figure 2—figure supplement 3*). These include 'female approaching', 'circling', 'head-butting', and 'female wing extension' (*Figure 2—figure supplement 3A–D*; *Videos 2* and *3*). We found that some of these behaviors were coupled; for example, 'circling' was often preceded by 'female shoving' (*Figure 2—figure supplement 3B*, inset and *Video 3*) and 'female wing extension' was often coincident with 'female chasing' (*Figure 2—figure supplement 3E* and *Video 2*), similar to male behavior during courtship (although we did not observe sounds from the females that resembled male courtship song (*Figure 2—figure supplement 3D*)). Our automated behavioral classifier did not find these behaviors because we only tracked the head and thorax of each fly, which did not provide enough information to automatically identify these behaviors, or to keep accurate track of identities during behaviors in which the male and female often overlap (e.g., during 'circling').

In sum, we found that for minutes following pC1-Int activation, females produced a variety of behaviors directed at the male. Some of these appear aggressive, such as shoving and head-butting (*Nilsen et al., 2004*; *Palavicino-Maggio et al., 2019*; *Schretter et al., 2020*), while others resemble

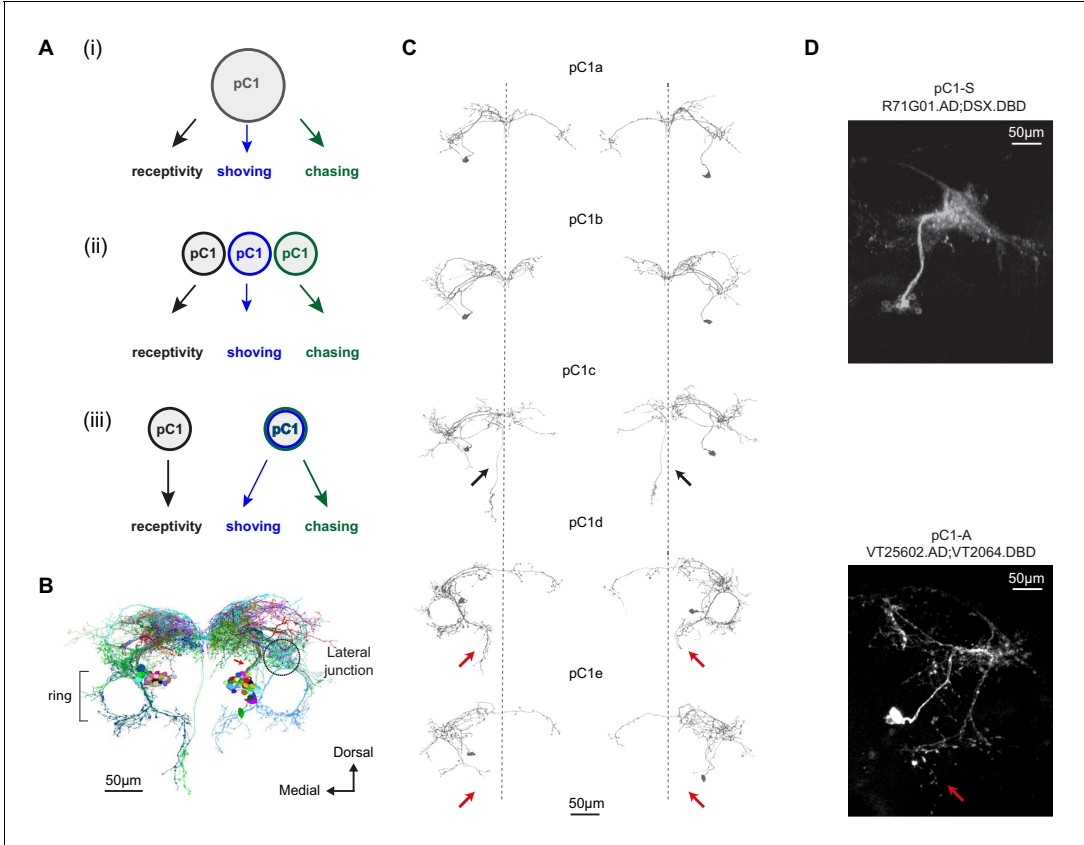

**Figure 3.** Defining pC1 cell types. (**A**) Models of how pC1 cell types control female receptivity, shoving, and chasing. In (i), a homogenous pC1 population drives three distinct behaviors, and in (ii) and (iii) pC1 is a heterogenous group, with different behaviors controlled by different pC1 subsets. (**B**) EM reconstruction of pC1 cells and other example neurons that pass through a cross section in the pC1 bundle. See *Figure 3—figure supplement 1* for all the cells that pass through the pC1 bundle, including neurons that project to the lateral junction (considered pC1 cells), and neurons that do not project to the junction (not considered pC1 cells). (**C**) Five pC1 cell types identified in FlyWire, mostly consistent with (*Wang et al., 2020a*). The medial projection in pC1c (see black arrows) is ipsilateral to the soma in one cell and contralateral in the other, likely reflecting variability between individual cells. Both pC1d and pC1e share a horizontal projection from the ring, but only pC1d cells have an extra vertical projection (red arrows indicate difference between pC1d and pC1e projections). The right pC1e cell is also missing part of the ring. (**D**) (top) Split GAL4 line pC1-S (R71G01. AD∩DSX.DBD; n = 8, 7.7 ± 5 cells per hemisphere) and (bottom) Split GAL4 line pC1-A (VT25602.AD∩VT2064.DBD; n = 7, 2 ± 0 cells per hemisphere). Neurons express GFP and are labeled with anti-GFP, see Key Resources Table for full genotype. pC1-A has a medial projection (red arrow), similar to pC1d neurons found in EM (**C**); the medial projection was found in 7/7 imaged pC1-A female brains, in both hemispheres. The pC1d and pC1e projections were not found in pC1-S imaged female brains (8/8 brains). The medial projection that is unique to the pC1c subtype was missing in all the 8 pC1-S flies we imaged - therefore, it is likely that pC1-S includes only pC1a and/or pC1b cells.

The online version of this article includes the following figure supplement(s) for figure 3:

**Figure supplement 1.** pC1 cells were identified in FlyWire based on morphology.

**Figure supplement 2.** Expression of driver lines pC1-Int, pC1-A, and pC1-S.

male courtship behaviors, such as chasing and unilateral wing extension (albeit without song). These behaviors typically peaked in the d3 condition, where they remained high after male introduction. In contrast, the effect on female receptivity and female responses to male song, were both strongest in the d0 condition, ruling out the possibility that the effect on female receptivity is an indirect consequence of modified male courtship behavior in response to changes in female behaviors. Below we identify the pC1 cell types that drive persistent shoving and chasing, determine that they are part of a recurrent neural network using a new EM connectomic resource, and demonstrate that they can drive persistent neural activity on timescales similar to behavior.

## pC1 cell types

We propose three possible circuit configurations to explain our behavioral results (*Figure 3A*). In the first configuration, the same set of pC1 neurons activate different downstream circuits, each one controlling a different behavior. The differences in the temporal dynamics arise downstream of pC1. In the second configuration, three non-overlapping subsets of pC1 neurons control the different behaviors. In the third configuration, one pC1 subset controls female receptivity (that peaks at d0), and another set controls chasing and shoving (both peaking at d3). The second and third models assume some functional heterogeneity in the pC1 population. To evaluate these circuit models, we examined the behavioral consequences of activating distinct subsets of pC1 neurons. To define pC1 cell types, we used automated reconstruction of neurons in an EM volume of a female brain (FAFB; *Zheng et al., 2018*); neuron segmentation and reconstruction were accomplished using a novel platform for visualization and proofreading called FlyWire (*Dorkenwald et al., 2020*). We examined the morphologies of neurons that send projections to the lateral junction through a thin neuronal bundle, as pC1 neurons are known to (*Figure 1A*, red arrow; *Figure 3B*, red arrow; see also *Deutsch et al., 2019* and *Zhou et al., 2014*).

We systematically checked all cell segments that pass through a cross-section in the pC1 bundle (*Figure 3—figure supplement 1A*) and excluded neurons that do not project to the lateral junction (*Figure 3—figure supplement 1B–D*), as all pC1 cells characterized so far project to the lateral junction (*Kimura et al., 2015*; *Rezával et al., 2016*; *Wu et al., 2019*; *Zhou et al., 2014*). We found five pC1 cells per hemisphere (*Supplementary file 1*), consistent with the cell types found from manual tracing in the same EM volume (*Wang et al., 2020a*), although with differences in some projections (*Figure 3C*; one cell/hemisphere for pC1a-e was also found in the hemibrain, a second EM dataset of the adult female brain [*Scheffer et al., 2020*]).

## The effects of pC1 subtypes on behavior

We used genetic intersections to label two non-overlapping pC1 subpopulations. pC1-A labels a single pC1d and a single pC1e neuron in each hemisphere (*Wu et al., 2019*; same as line pC1dSS3 [*Schretter et al., 2020*]) and no cells in the VNC (*Figure 3D* and *Figure 3—figure supplement 2*; $2 \pm 0$, n = 7). The second intersection (pC1-S, a split-Gal4 intersection between Dsx.DBD *Pavlou et al., 2016* and R71G01.AD; *Figure 3D* and Fly genotype table) does not label pC1d or pC1e cells, as evidenced by absence of the medial projections of pC1d/e neurons (*Figure 3C–D*). The pC1-S line labels $7.7 \pm 4$ pC1 cells per hemisphere in the brain, and no cells in the VNC (*Figure 3—figure supplement 2C*). All cells in the pC1-S line project to the lateral junction (*Figure 3D*).

Next, we tested if activation of these two non-overlapping pC1 sub-populations drives persistent behavioral phenotypes. Activation of either line pC1-A or pC1-S did not affect female receptivity (*Figure 4A,D*), but activation of pC1-A drove persistent shoving and chasing (*Figure 4B–C*), while activation of pC1-S did not (*Figure 4E*). These results are consistent with model 3 (*Figure 3A*), in which female receptivity and female shoving/chasing are driven by different populations, and also consistent with prior work, showing that 10 min of thermogenetic pC1-A activation drives persistent chasing in females (*Wu et al., 2019*). pC1d/e neurons are female-specific (*Wu et al., 2019*), and prior work reveals that pC1 neurons in females, but not males, respond to courtship song (*Deutsch et al., 2019*). Using in vivo whole-cell patch clamp recordings of neurons labeled in the pC1-A line, we found that pC1d/e neurons in virgin females depolarize in response to features in conspecific courtship song (*Figure 4—figure supplement 1*). This finding indicates that pC1d/e neurons can be activated during courtship by male song.

When examining the fraction of time spent shoving or chasing across the three conditions (control, d0, and d3) to compare results of activating neurons in pC1-A and pC1-Int lines (*Figure 4*), we found that levels of female chasing are increased with pC1-A activation (relative to pC1-Int) at d3. The levels of shoving are slightly decreased at d0, but this effect was not significant. This suggests that pC1 cell types in the pC1-Int driver, other than pC1d/e (and other than the cells in the pC1-S driver), have a modulatory effect on the shoving and chasing behaviors, at distinct timepoints.

pC1-Int, but not pC1-A, activation drives persistent female receptivity (*Figure 1F* vs *Figure 4A*), indicating that pC1-Int cells other than pC1d/e drive female receptivity (although high copulation rates in control flies could mask a potential effect of pC1-A on female receptivity). We hypothesized that the same pC1-Int cells that drive female receptivity, also affect the probability of shoving and

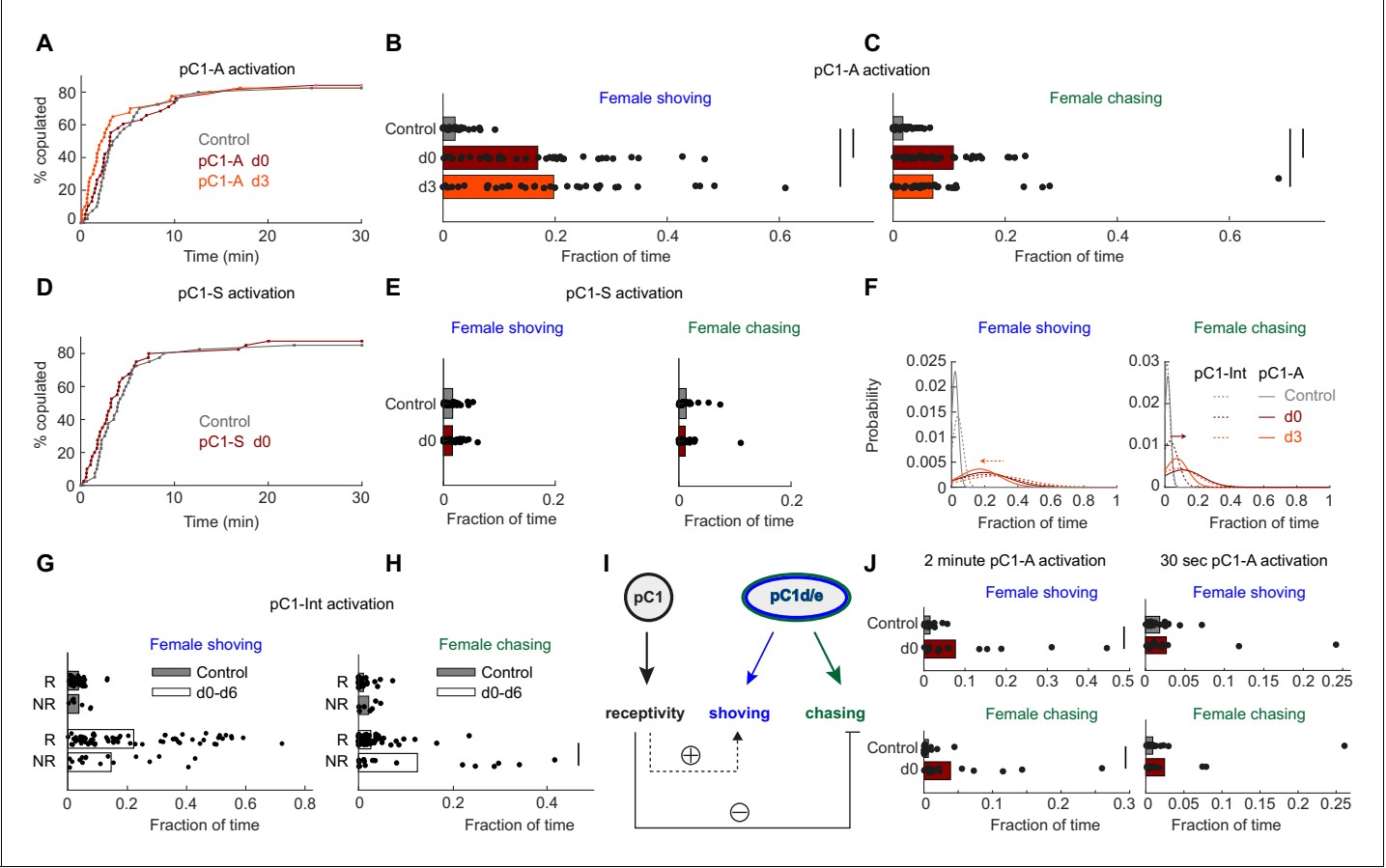

**Figure 4.** pC1d/e neurons drive female shoving and chasing, but do not affect receptivity. (A) pC1-A activation did not affect copulation rates in either the d0 or d3 conditions (n = 40, 38, 40 for control/d0/d3; p=0.79 or 0.29 for control vs d0 or d3; Cox proportional hazards regression, see Materials and methods). (B, C) Shoving (B) and chasing (C) probabilities (control/do/d3: 0.02/0.17/0.20 and 0.018/0.11/0.07 for shoving and chasing) were significantly higher in both the d0 and d3 conditions compared to control (two-sample t-test; *p<0.05). (D–E) same as (A–C), but for pC1-S activation. pC1-S activation did not affect either copulation rate (D) or shoving or chasing (E) probabilities (control/do: 0.02/0.02 and 0.01/0.01 for shoving and chasing). (F) Probability distribution of the fraction of time the female spent shoving (left) or chasing (right) following pC1-A activation (solid line) or pC1-Int activation (dashed line). Arrows indicate the difference in peak shoving (at d3; two-sample t-test, p=0.11) or chasing (at d0; two-sample t-test, *p=4.5*10$^{-4}$) probability, between pC1-Int and pC1-A activation. Bonferroni correction was used for multiple comparisons. (G) Fraction of frames with shoving for pairs in which females copulated (R - Receptive) or did not copulate (NR – Not Receptive) for all experimental conditions taken together (d0–d6). Each dot is a single pair, and the bar value is the mean over all pairs (p=0.92 and 0.13 for control and d0-d6). (H) Same as (G), for female chasing (p=0.13 and p<10$^{-5}$ for control and d0-d6). (I) Female pC1d/e cells drive persistent shoving and chasing, but do not affect female receptivity. Female receptivity, controlled by a separate pC1 subset labeled in the pC1-Int line, suppresses female chasing, while possibly enhancing female shoving. (J) Shoving and chasing probabilities (control/d0: 0.077/0.015 and 0.039/0.006 for shoving and chasing; p=0.026 and 0.035; n = 20/20 for control/d0) were significantly higher in the d0 condition compared to control following 2 min of pC1-A activation, but not following 30 s (n = 20/21 for control/d0) of pC1-A activation (p=0.52 and p=0.24 for shoving and chasing).

The online version of this article includes the following figure supplement(s) for figure 4:

**Figure supplement 1.** Auditory responses in pC1d/e neurons recorded via whole-cell patch clamp.

chasing. If true, we would expect a different rate of shoving and chasing in receptive and non-receptive females. Indeed, we found that following pC1-Int activation, females that eventually copulated (receptive) showed a slightly higher level of shoving compared with females that did not eventually copulate (unreceptive), although this effect was not statistically significant (*Figure 4G*). In contrast, receptive females showed a strong and significant reduction in chasing (*Figure 4H*). Taken together, our results suggest that cells within the pC1-Int line that control receptivity modulate the amount of female chasing (and possibly also shoving) (*Figure 4I*). We also tested shorter activation durations and found robust persistent shoving and chasing following 2 min activation, but reduced persistent

shoving and chasing following 30 s activation (*Figure 4J*). Below we address how minutes-long pC1d/e activation affects neural activity.

## pC1d is reciprocally connected to a specific subset of Fruitless+ neurons

The pC1-A line (same as pC1dSS3 [*Schretter et al., 2020*]) includes two cells per hemisphere, one pC1d and one pC1e cell. *Schretter et al., 2020* have demonstarted that optogenetic activation of genetic lines that contain pC1d, but not pC1e, drive female aggressive behaviors, such as those we refer to here as 'shoving'. We therefore first focused on pC1d, and mapped the major inputs and outputs, searching for circuit motifs that could account for pC1d's ability to drive a persistent behavioral state in females. We used automated reconstruction of all neurons within an EM volume of an entire adult female brain called FAFB (*Zheng et al., 2018*; *Dorkenwald et al., 2020*). This volume contains both brain hemispheres, enabling complete reconstruction of pC1d cells that send projections across the midline (*Figure 3C*). Focusing on a single pC1d cell (*Figure 5A*) we first manually detected synaptic connections (*Figure 5—figure supplement 1A*; see Materials and methods), as done previously for other circuits in FAFB (*Felsenberg et al., 2018*; *Sayin et al., 2019*; *Zheng et al., 2018*) - while our manual detection process was unbiased, in that we looked for synapses between pC1d and any other overlapping segment in FlyWire (see Materials and methods), we only sampled a subset of all pC1d synapses (see below for analysis based on automated synapse detection). After proofreading the pC1d cell and its input and output cells, and excluding neurons with weak connections (using three synapses as a threshold, see Materials and methods), we counted a total of 417 presynaptic and 421 postsynaptic sites (*Figure 5B*, *Video 4*). We sorted all pC1d synaptic partners by cell type, based on morphology, and examined the distribution of synapses by type for input (presynaptic partners) and output (postsynaptic partners) neurons separately (*Figure 5C*). The three output types with the largest number of synapses with pC1d share a common morphology: all pass through a single neurite bundle (*Figure 5D–E*, *Figure 5—figure supplement 1B–C* and *Video 5*) and all send projections to the ring (*Figure 5A*), including dense projections in the lateral junction (*Figure 5E*).

The top matches (using NBLAST [*Costa et al., 2016*]) for all three types were sexually dimorphic Fru+ neurons called aIP-g (*Figure 5E* and *Figure 5—figure supplement 1B*, see *Cachero et al., 2010*). FlyWire cells that share the aIP-g morphology were sorted into three types, aIPg-a, aIPg-b, and aIPg-c, based on the three separate bundles through which their projections pass (*Figure 5E*). According to our manual synapse detection, aIPg-a cells have 131 sites postsynaptic to pC1d and only five presynaptic sites, while aIPg-b,c cells have stronger reciprocal connections with pC1d, with an output:input ratio of ~1:1 (38:39) for aIPg-b, and 2.8:1 (39:14) for aIPg-c.

FlyWire provides a mapping of the publicly available, automatically detected synapses from *Buhmann et al., 2019*; see Materials and methods for details. We re-evaluated synaptic partners using automatic detection, and focused on cells with strong connections with pC1d using two criteria: (1) minimum of six synapses, (2) the cell belongs to a cell type (based on morphology) with at least one cell with 15 synapses or more with pC1d. Consistent with manual detection, we found that aIPg-a cells are postsynaptic to pC1d, while aIPg-b,c are reciprocally connected to pC1d (*Figure 5F*). Some cells from the aIPg-b,c groups are also interconnected (*Figure 5G*). Interestingly, the most interconnected cells within the aIPg-b group were also the ones that are reciprocally connected to pC1d (*Figure 5G*, red lines).

We also examined synaptic connectivity between pC1d and aIPg cells in a second EM database that consists of a portion of the adult female brain (the 'hemibrain' [*Scheffer et al., 2020*]), and found a set of 13 neurons identified as aIPg (also evaluated in [*Schretter et al., 2020*]), compared with 39 neurons we identified as aIPg in FlyWire. Twelve of these aIPg cells (denoted as types aIPg1-3 in the hemibrain) share the aIPg-b morphology (*Figure 5—figure supplement 1D*), and are synaptically connected to pC1d (excluding one connection with less than three synapses). One (aIPg4) shares the aIPg-c morphology. Consistent with our results in FlyWire, pC1d in the hemibrain has more presynaptic sites than postsynaptic sites with aIPg-b (hemibrain aIPg1-3) cells, and aIPg-b neurons form many recurrent connections with each other (*Figure 5H*). Note that while our classification of aIPg cell types was based on morphology alone, the classification in the hemibrain is based on both morphology and connectivity. In the hemibrain v1.1, we found additional cells that match

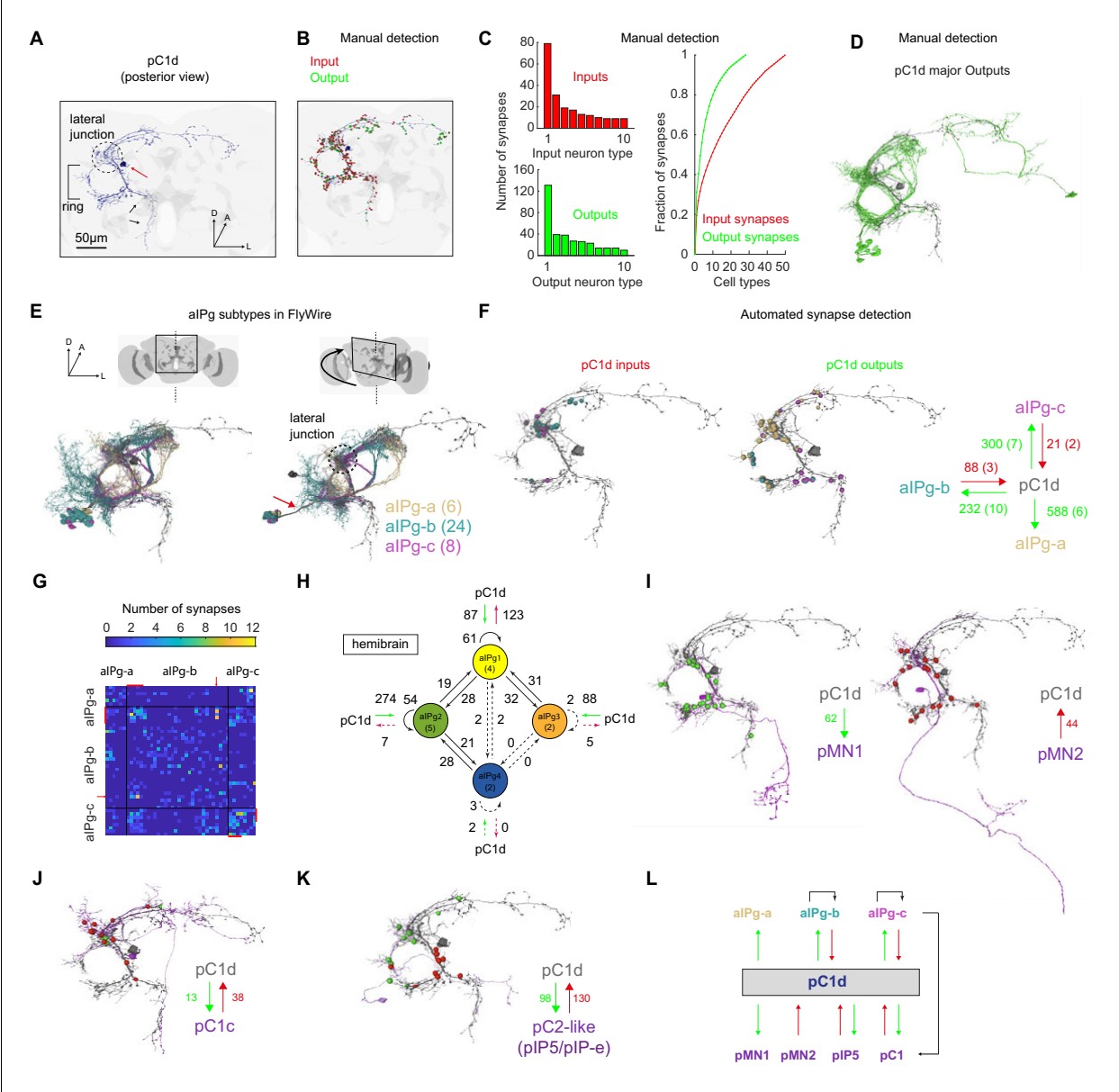

**Figure 5.** The connectome of pC1d reveals recurrent connections with aIPg neurons. (**A**) The left pC1d cell (posterior view) reconstructed in FlyWire following automated segmentation and manual proofreading. The cell body is marked with a red arrow, and the pC1d long medial projection (that does not exist in other pC1 types) is marked with black arrows. (**B**) Same cell as in (**A**), with manually detected synapses. Neurons pre-synaptic (inputs to pC1d) are marked in red, and neurons post-synaptic (outputs of pC1d) in green. After excluding segments connected to pC1d with less than three synapses, we counted 417/421 manually-detected input/output synapses (see also *Video 4*). (**C**) Left: pC1d inputs (66 cells) and outputs (50 cells) were classified into cell types based on morphology. The number of input (top) or output (bottom) synapses are shown for each cell type, sorted (separately for inputs and outputs) based on the total number of synapses with pC1d for each type. Right: The cumulative fraction of synapses counted as a function of the number of cell types included (calculated separately for inputs/outputs). The three most common output types encompass 49.4% of all output synapses, while the three most common input types encompass 30.5% of all input synapses. (**D**) The cells that belong to the most common cell types (50% of all output synapses) are shown for pC1d output cells. Note that pC1d has postsynaptic connections with both left (ipsilateral to cells body) and right (contralateral) aIPg cells. (**E**) Left: Posterior view (same as in (**A**)) of pC1d (gray) and aIPg cells. Right: rotated view, showing the separation between three subtypes of aIPg cells (6/24/8 for aIPg-a/b/c), sorted according to their projections. See Table 1 for the full list of FlyWire detected aIPg cells in one hemisphere. (**F**) Synapses between pC1d (gray) and individual example aIPg cells, color coded by aIPg type as in (**E**). Total synapse count for each type is summarized. Synapses were detected automatically (*Buhmann et al., 2019*). Only cells with six synapses or more with pC1d are included in the analysis (See Materials and methods and Table 1 for more details). (**G**) Connection matrix between pairs of aIPg cells. The number of synapses (*Buhmann et al., 2019*) between a given pair of aIPg cells are indicated with a colorscale. Black lines separate aIPg subtypes. Red lines denote the aIPg cells that are reciprocally connected with pC1d (with si synapses or more each way), and red arrows indicate an

*Figure 5 continued on next page*

*Figure 5 continued*

aIPg-b cell that is reciprocally connected to cells that form reciprocal connections with pC1d. (H) The number of synapses within and between groups of aIPg cells based on the fly hemibrain connectome (*Scheffer et al., 2020*). The number in parentheses indicates the number of cells per group (aIPg-1–4). Round arrows indicate within-group connections (e.g. 61 synaptic connections between pairs of aIPg-1 cells). Dotted arrows are shown for weak connections (under five synapses). (I) pC1d connections with Dsx+ pMN1 and Dsx+ pMN2 cells. (J) An example Dsx+pC1 cell (type pC1c) that is recurrently connected to pC1d. (K) pC1d connections with pC2-like cells (with similar morphology as hemibrain pIP5 neurons or Fru+ pIP-e clones from *Cachero et al., 2010*). (L) pC1d is a hub connecting Dsx+ or Fru+ pC1, pIP5, pMN1, and pMN2 neurons with Fru+ aIPg neurons.

The online version of this article includes the following figure supplement(s) for figure 5:

**Figure supplement 1.** Comparison of aIPg cells identified in FlyWire and Hemibrain.

(morphologically) neurons we term aIPg-a and aIPg-c – these neurons are called SMP555/556 and SMP558, respectively.

Finally, in FlyWire, we found that pC1d forms connections with other Dsx+ cells, including direct connections with pMN1 (DNp13) and pMN2 (vpoDN) (*Figure 5I*), other pC1 cells (*Figure 5J*), and pC2-like cells with similar morphology to Fru+ pIP5 cells (*Figure 5K*, *Video 6* and *Supplementary file 1*). These results indicate that pC1d may serve as a hub within the central brain for Dsx+ and Fru+ neurons (*Figure 5L*).

Using automated synapse detection in FlyWire, we examined all the major inputs and outputs of both pC1d (*Figure 6A*) and pC1e (*Figure 6—figure supplement 1*), as our driver line pC1-A labeled both neurons in each hemisphere (see full list of inputs and outputs and public links to FlyWire neurons in *Supplementary file 1*). aIPg-b neurons have the most reciprocal connections with pC1d, and the three strongest output types of pC1d are aIPg-a (588 synapses), aIPg-c (300 synapses), and aIPg-b (232 synapses). The pC1d output:input ratio with aIPg (pC1d-to-aIPg:aIPg-to-pC1d) was 588:0 for aIPg-a, 2.6:1 for aIPg-b and 14.3:1 for aIPg-c (*Figure 5F*) – consistent with the results obtained by manual detection, showing that aIPg-b has the most 'balanced' reciprocal connectivity with pC1d. For pC1e (*Figure 6—figure supplement 1*), of the top inputs and outputs, the only reciprocal connections were with aIPg-b. Shared inputs between pC1d and pC1e also include pC1a, a cell previously shown to control female receptivity (*Wang et al., 2020b*). Shared outputs include aIPg-a,c, and cell types that share the hemibrain morphology of SIP024 and LAL003. Some of the top connections of pC1d were cross-hemispheric neurons or neurons with synapses contralateral to pC1d (*Figure 6B*) – these neurons and connections fall outside of the volume of the hemibrain. However, several of the top inputs and outputs could be identified in the hemibrain, and we have indicated the corresponding cell type name where we could find matches (*Figure 6* and *Figure 6—figure supplement 1*). Seven of the aIPg cells that are pC1d synaptic partners, also synapse with pC1e (*Figure 6—figure supplement 1*).

As recurrent connectivity between neurons is known to support persistent neural activity (*Goldman-Rakic, 1995*; *Major and Tank, 2004*;

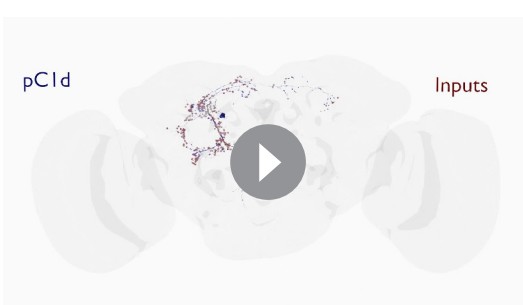

**Video 4.** A single pC1d neuron, automatically traced and manually proofread. Inputs (post-synaptic terminals, manual detection) are shown in red, outputs (pre-synaptic terminals, manual detection) in green (see also *Figure 5A–B*).
https://elifesciences.org/articles/59502#video4

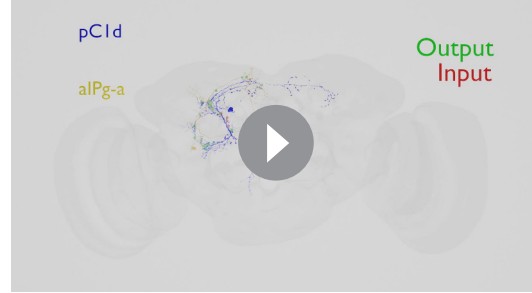

**Video 5.** pCd (blue) is shown with example aIPg-a,b,c cells. Synapses (detected manually) are marked in red for inputs (to pC1d) and in green for outputs. Cell type colors (yellow, cyan, magenta) are shown for aIPg-a,b,c as in *Figure 5E–F* and *Figure 5—figure supplement 1*.
https://elifesciences.org/articles/59502#video5

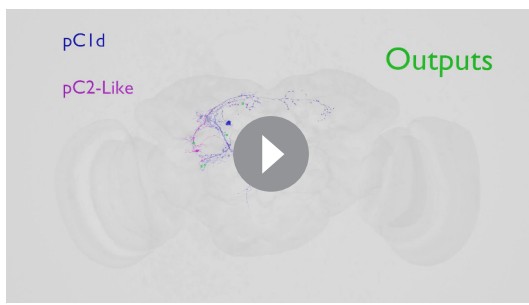

**Video 6.** pC1d (blue) is shown with neurons that have a similar morphology as known female Doublesex-expressing cells. pC1 subtypes are shown in *Figure 3C* and two pC2 subtypes are shown in *Figure 5K*. pC1d input/output synapses (manual detection) with each example cell are shown in red/green.
https://elifesciences.org/articles/59502#video6

*Zylberberg and Strowbridge, 2017*), we next examined whether activating pC1d/e could drive long-lasting changes in brain activity, with a spatial distribution that matches the pC1d or pC1e connectomes.

## Activation of pC1d/e neurons drives persistent neural activity

Persistent neural activity is defined as activity that continues after a triggering stimulus ends (*Zylberberg and Strowbridge, 2017*). To relate our findings above to persistent neural activity, we first activated either neurons in the pC1-A or pC1-S lines using 5 min of optogenetic stimulation (similar to behavioral experiments, see Materials and methods), and imaged responding cells via GCaMP6s expressed pan-neuronally (*Figure 7A*). To compare activity across flies and to map activity onto a reference atlas, we used a recently developed pipeline for two-photon volumetric calcium imaging, motion correction, registration, and region-of-interest (ROI) segmentation (*Pacheco et al., 2021*), and scanned the entirety (in the anterior-posterior axis) of the dorsal half of a single brain hemisphere (either left or right) in each fly (*Figure 7A*) – we mirrored all activity onto one hemisphere for display. We measured neuronal activity during the 5 min of optogenetic activation in addition to 9.5 min following activation offset (*Video 7*), and found that out of 47,882 ROIs with activity segmented across 28 brains and all genotypes (*Figure 7B*; see Materials and methods), 4254 ROIs showed significant responses to optogenetic stimulation (*Figure 7C*; Ft1 > 3$\sigma_0$, $\sigma_0$ = standard deviation of activity during baseline).

We then clustered these ROIs based on response patterns (*Figure 7D*), which revealed four types of responses. Transient responses - ROIs with elevated activity during the optogenetic stimulus (t1), but not following the stimulus (t2) - could be grouped into two clusters (response types 3 and 4). The other two types showed sustained activity lasting at least 5 min after the optogenetic stimulus offset (response types 1 and 2; Ft2 > 3$\sigma_0$, see Materials and methods). The temporal dynamics of persistent neural activity, continuing to at least 10 min following stimulation, is consistent with our observation of female shoving and chasing of a male introduced 6 min after stimulation offset (*Figure 2E–F*). In addition, response type 2 could be split into two clusters (*Figure 6D*, right) based on response temporal dynamics.

While response type 1 had low spatial consistency across animals, response types 2–4 showed higher spatial consistency, and the spatial distribution of ROIs differed between controls, pC1-S, and pC1-A activated flies (*Figure 7E*). Activation of pC1d/e neurons (in line pC1-A) drove persistent activity (response type 2) in more than 30% of the imaged flies, and in 24.7 times more voxels than in controls, and 6.8 times more voxels compared with pC1-S activation. Making use of neuropil segmentation of an in vivo brain atlas to which all ROIs were registered (*Pacheco et al., 2021*), we evaluated the distribution of pC1d/e-elicited activity by brain neuropil (*Ito et al., 2014*). Persistent activity (response type 2) was clustered in the posterior-dorsal portion of the brain spanning the Superior Medial, Lateral and Intermediate Protocerebrum (SMP, SLP, SIP), the Anterior Optic Tubercle (AOTU), and the Inferior and Superior Clamp (ICL and SCL; *Figure 7F* and *Figure 7—figure supplement 1A*); these brain regions contain a large number of projections from sexually dimorphic neurons expressing either Doublesex or Fruitless (*Rideout et al., 2010*; *Yu et al., 2010*). We found that 61% (4717/7759) of all the presynaptic terminals and 85% (3283/3848) of all the postsynaptic terminals for the group of 11 aIPg1-3 (all share the aIPg-b morphology) cells in hemibrain v1.1 are in these six areas, with the SMP, SIP, and AOTU being the most dominant input and output regions for the aIPg1-3 cells.

Our behavioral results indicated that female brain state must differ between the d0, d3, and d6 conditions (*Figure 2F* and *Figure 4B–C*) – we therefore quantified neural activity at these specific time points (0, 3 min, and 6 min) following optogenetic activation. In the neuropils with highest

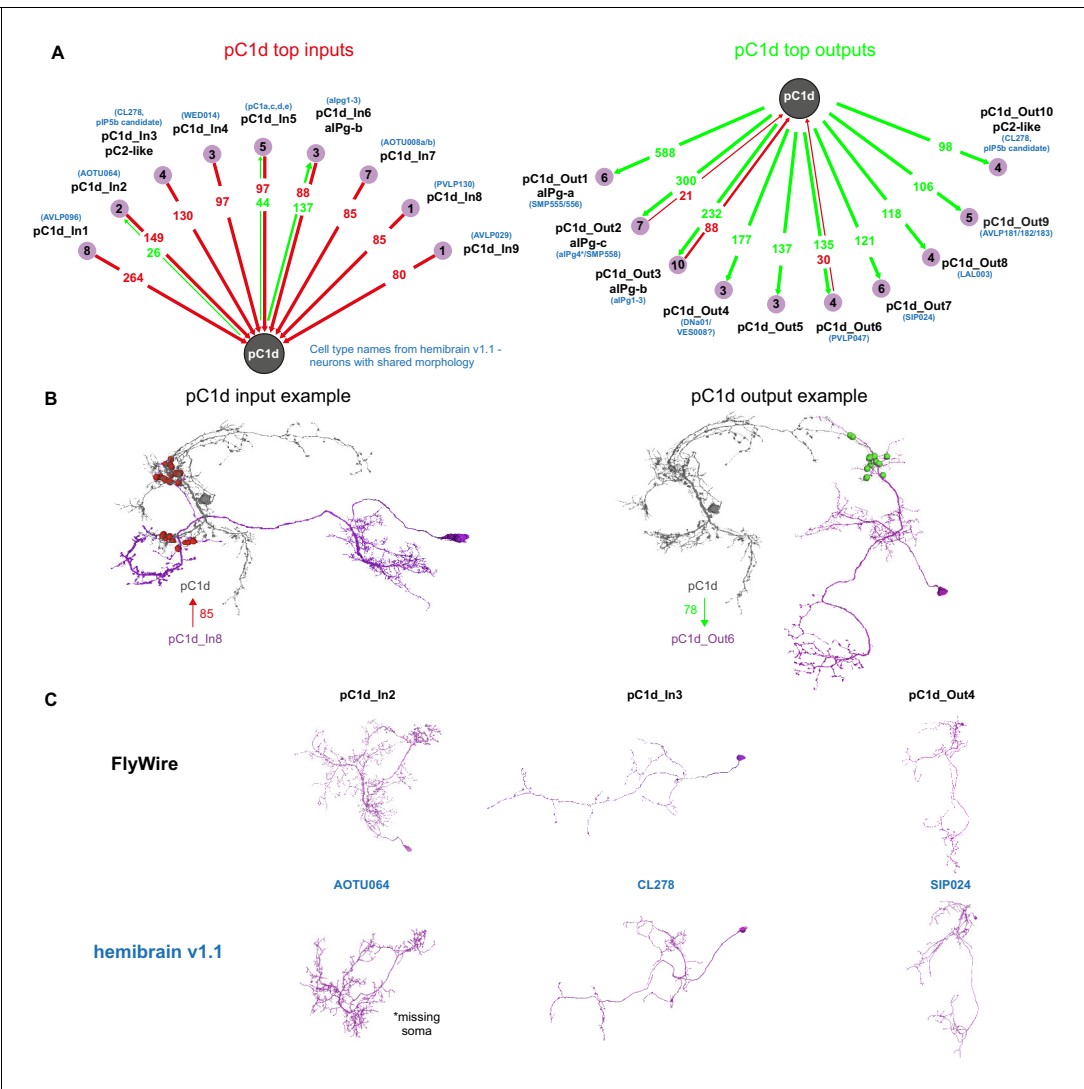

**Figure 6.** pC1d synaptic partners based on automated synapse detection in FlyWire. (**A**) The top (≥80 synapses) pC1d inputs (left) and outputs (right). The number of pC1d presynaptic (input) and postsynaptic (output) connections are shown in red/green. The number of input or output cells that are connected to pC1d with six synapses or more (see Materials and methods) are indicated in the purple circles. Cells with shared morphology were identified in hemibrain v1.1 (names shown in blue) for each cell type. As the number of synapses is counted per type and only for cells with six synapses or more, separately for the inputs and outputs, the numbers between the left and right plots do not necessarily match (see Table 1 for more details). (**B**) Example cells that are connected to pC1d. pC1d_In8 synapses onto pC1d on a branch ipsilateral to pC1d soma, but pC1d_In8 has its soma and dendrites in the contralateral hemisphere. pC1d_Out6 connects to pC1d on a branch contralateral to the pC1d soma. (**C**) Example pC1d input or output cells in FlyWire, compared to similar cells in the hemibrain v1.1.

The online version of this article includes the following figure supplement(s) for figure 6:

**Figure supplement 1.** pC1e synaptic partners based on automated synapse detection in FlyWire.

activity, we found that responses were highest immediately following stimulation (t = 0) and decayed significantly by t = 3 min, and further still by t = 6 min (*Figure 7G*; *Figure 7—figure supplement 1B*).

In order to measure the overlap of pC1-elicited activity with Dsx+ neurons, we generated anatomical labels for the lateral protocerebral complex (LPC), a diffuse brain area to which all Dsx+ neurons send their projections, and also for all major groups of Dsx+ somas (pC1, pC2l, pCd1, and pCd2) within the in vivo brain atlas (*Figure 7F*, see Materials and methods). We found that ROIs with persistent activity (response type 2) overlap with the LPC, in addition to the regions occupied by pC1 somas, and to a lesser extent with regions occupied by pC2, pCd1, and pCd2 somas,

suggesting that Dsx+ pC1 neurons carry persistent activity. We also looked for overlap between ROIs with persistent activity and the projections of individual aIPg neurons, all registered into the same reference brain (*Figure 7H*; see Materials and methods). While persistent activity (response type 2; *Figure 7D–E*) spans only 4.3% of the central brain (*Figure 7E*; see Materials and methods), we found response type 2 activity in 20.14% (union of the overlap) of the voxels that include aIPg-a/b/c example cell (one of each type) from FlyWire (*Figure 7H*). In addition, activity driven by 5 min of activation was not aberrant. DF/F values following 5 min optogenetic activation (data from *Figure 7C*) fell within the distribution of DF/F values observed in separate experiments (*Pacheco et al., 2021*) in which activity was driven by auditory stimuli rather than optogenetic activation (*Figure 7—figure supplement 1C*).

We also examined persistent activity following a shorter activation period of 2 min, and found ROIs with persistent activity (response Type 2; *Figure 7I–J*). The persistent activity following 2 min activation was weaker in this condition compared to 5 min activation, suggesting that persistent activity scales with the activation period.

We next expressed GCaMP6s in only Dsx+ neurons, to confirm the specific Dsx+ cells with persistent neural activity – this is possible because Dsx+ somas are clustered by cell type (*Figure 1A*). We activated pC1d/e neurons for 5 min (*Figure 8A* and *Video 8*) and recorded activity in 273 cells (ROIs drawn manually) across 16 flies. We examined the responses during (t1) and after (t2) optogenetic stimulation (same as for the pan-neuronal dataset), and compared these responses to controls in which pC1d/e neurons were not activated (n = 11 flies, 192 ROIs; See Fly genotype table for full genotypes; *Figure 8B*). A number of Dsx+ pC1 cells showed strong persistent activity (*Figure 8B*; same definition as for the pan-neuronal screening, $Ft2 > 3\sigma_0$) following optogenetic activation. We observed some heterogeneity in responses across the pC1 cells (*Figure 8B*), with some cells showing faster decay than others following stimulus offset, consistent with the two clusters underlying response type 2 (*Figure 7D*, green and brown). We did not observe persistent activity in any non-pC1 Dsx-expressing cell types (*Figure 8—figure supplement 1*), including pC2 neurons or pCd1 neurons, previously shown to be necessary for P1-induced persistent activity in males (*Jung et al., 2020*; *Zhang et al., 2019*).

Last, using the same methodology, we examined neural activity in single Fru+ cells following pC1d/e activation, by expressing GCaMP6s via the Fru-LexA driver (*Figure 8C*, top). We found persistent activity in two group of cells, denoted as 'Group 1' and 'Group 2' (*Figure 8C*, bottom), that often lasted over a minute following activation (*Figure 8D–E*) – ROIs drawn manually for individual cells within each group. By comparing the location of Fru+ cell bodies with persistent activity with the position of single Fru+ cells and to the location of pC1/pC2/aIPg cells in FlyWire (*Figure 8—figure supplement 2*). We conclude that Group 2 includes pC1 neurons, while Group 1 likely includes pC2/pIP5 neurons and possibly also aIP-g cells. The persistent activity is most likely not in Dsx+/Fru+ pC2 neurons given our observations that Dsx+ pC2 cells do not show persistent activity following pC1d/e activation, but could be in pIP-e (also called pIP5 in the hemibrain). This is consistent with our analysis in FlyWire, showing that pIP5 is both pre- and post-synaptic to pC1d.

In sum, our pan-neuronal imaging reveals that female brain state is different at 0 and 3 min following activation, providing an explanation for the differences in behaviors produced following introduction of a male at these different delays. In addition, by clustering response types, we were able to map pC1d/e-driven persistent neural activity to brain regions containing both Dsx+ neurons and Fru+ aIPg neurons, and with follow-up experiments showed that several Dsx+ pC1 neurons as well as Fru+ putative pC1 and aIPg cells contain persistent neural activity. This is consistent with the recurrent circuit architecture we found using EM reconstruction (*Figures 5–6*).

## Discussion

We find that pC1 neurons drive a persistent internal state in the *Drosophila* female brain that modulates multiple behaviors over timescales of minutes (receptivity, responses to male courtship song, aggressive behaviors, and male-like courtship behaviors [*Figures 1–2* and *4*]). The behavioral effects we observe may be similar to the effects of 'emotion states' observed in other animals, such as mice, fish, and primates (*Anderson and Adolphs, 2014*; *Kunwar et al., 2015*; *Posner et al., 2005*; *Russell, 2003*; *Woods et al., 2014*). In general, effects on behavior of such emotion states are thought to scale with levels of persistent neural activity (*Hoopfer et al., 2015*; *Lee et al., 2014*). We found

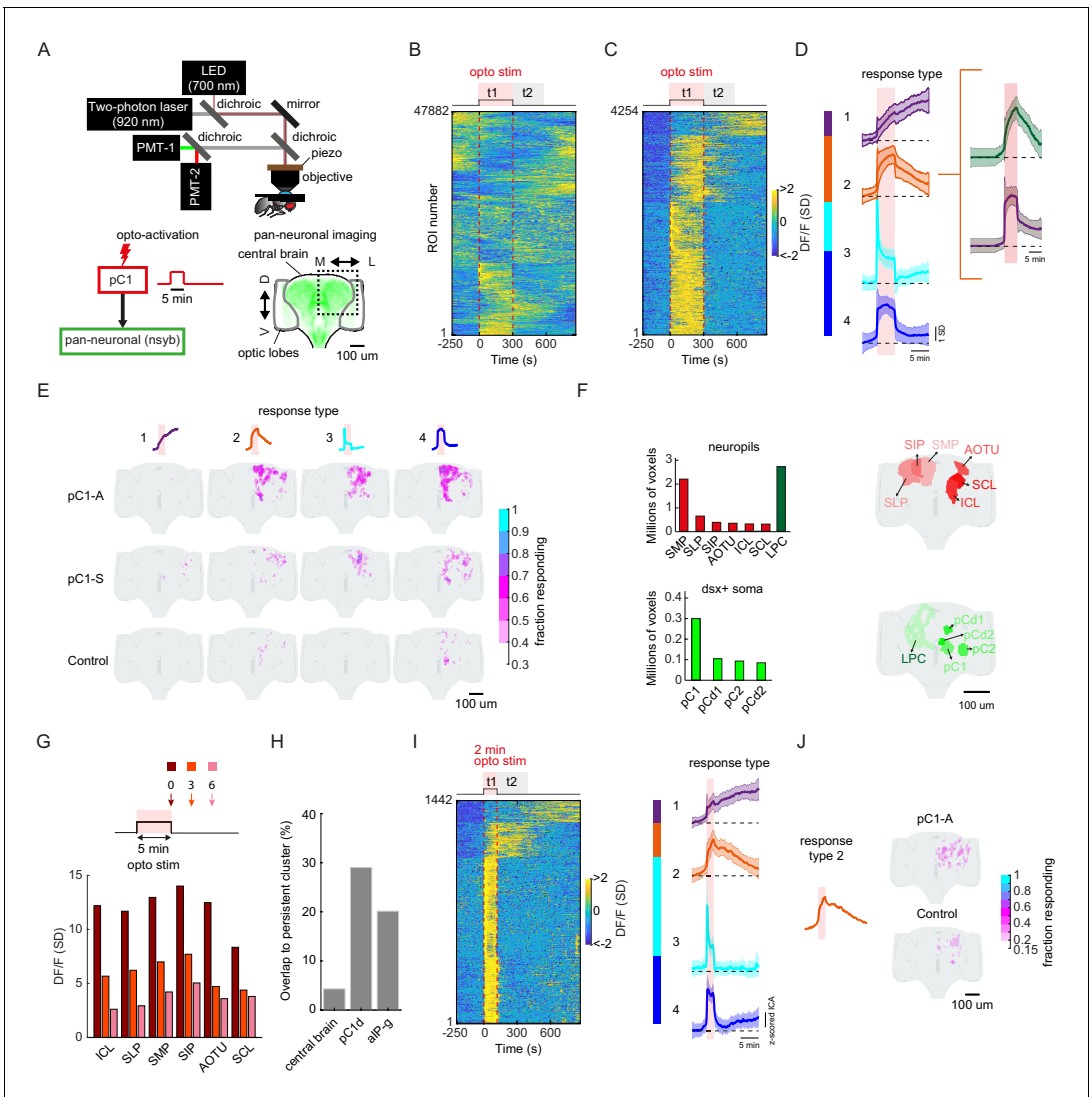

**Figure 7.** pC1d/e neurons drive persistent neural activity in the central brain. (**A**) Experimental setup. pC1 cells (pC1-A or pC1-S) expressing csChrimson were activated through the objective using an LED (700 nm). GCaMP6s and tdTomato were expressed pan-neuronally using the nsyb driver, and a custom-designed two-photon microscope was used to image brain activity before, during and after pC1 activation (see Key Resources Table for genotypes and Materials and methods for more details on the experimental setup). (**B**) Brain activity recorded in response to optogenetic stimuli (n = 28 flies, all genotypes). GCaMP6s signal was motion corrected and 3D-ROI segmented based on correlated activity in neighboring voxels (see Materials and methods). The z-scored signal of all ROIs (n = 47882 ROIs from both pC1-S and pC1-A activation and control experiments) are plotted in units of standard deviations (see scale bar), and shown 5 min before activation, during activation (t1), and 9.5 min post-activation (t2 marks the first 5 min post-activation). Red dashed line depicts the optogenetic stimulus onset and offset. (**C**) pC1 activation evokes both transient and persistent activity. Subset of ROIs from panel (**B**) were selected based on mean z-scored activity during (t1) and after photoactivation (t2), and ROIs were sorted by hierarchical clustering of temporal dynamics. We found 4254 responsive ROIs, defined as ROIs with Ft1 > 3$\sigma_o$ ($\sigma_o$ - standard deviation during baseline, Ft1 is the mean fluorescence during t1), including transient (Ft2 $\leq$ 3$\sigma_o$, blue and cyan; Ft2 is the mean fluorescence during t2) or persistent (Ft2 >3$\sigma_o$, orange and purple) response types . (**D**) Mean ± SD for response types 1–4. In response types 1 and 2, the activity level (calcium response) persists after activation offset, while for types 3 and 4, the activity is high during, but not after photoactivation. The two major sub-clusters of response type 2 are shown at right. (**E**) Maps of transient and persistent activity types. ROIs from response types 1–4 per animal were registered to an in vivo intersex atlas (**Pacheco et al., 2021**) to generate probability density maps across animals per brain voxel (each voxel is 0.75 × 0.75 × 1 μm³). Activity maps are maximum-projected along the anterior-posterior axis, and overlaid onto the brain template, color coded by the fraction of flies showing activity at each voxel (ranging from 30% to 100%). We considered a voxel to consistently have a particular response type if active in over 30% of flies. Response type 2 shows persistent activity following pC1-A activation, and occupies 4.3% of the volume imaged, compared with 0.6% following pC1-S and 0.2% in control flies. (**F**) Brain regions containing persistent responses (type 2). We used both anatomical segmentation of the in vivo brain atlas (**Pacheco et al., 2021**) and segmentation of the Dsx+ circuit (also registered to the same atlas) into processes in the LPC and major groups of cell bodies (pC1, pC2, pCd1, pCd2) to assign ROIs to neuropils (red) or overlap with Dsx+ neurons (green). For each of these regions, we calculated the

*Figure 7 continued on next page*

*Figure 7 continued*

average number of voxels or volume (across-individuals) occupied by all ROIs belonging to response type 2, following pC1-A activation. Neuropils were sorted by the number of voxels, and the top six neuropils are shown. pC2m and pC2l are shown together as pC2, as they are not always spatially separable in females. For responses in other conditions (pC1-S, control) and other neuropils see *Figure 7—figure supplement 1A*. (G) Mean response (DF/F) over all flies and ROIs per brain neuropil from (F), at t = 0 (stimulus offset), t = 3 min and t = 6 min. Time points relative to stimulus are shown in arrows in the schematic. Each ROI's activity was z-scored relative to the baseline; therefore, DF/F units are plotted in standard deviation (SD) relative to baseline activity. (H) The percent of voxels that belong to the persistent cluster (response type 2) out of the volume imaged in the central brain (4.3%), out of the voxels that include pC1d (29.04%), or out of the voxels that include aIPg cells (20.14%); see Materials and methods – pC1d and aIPg neurons from FlyWire were registered into the in vivo atlas for comparisons. (I) Shorter duration (2 min) pC1-A activation also evokes both transient and persistent activity. ROIs for both control and pC1-A activation (using the same criteria as in (C)) could also be clustered into four response types (purple, orange, blue, and cyan) similar to (D). (J) Map of persistent activity type 2 upon 2 min pC1-A stimulation.

The online version of this article includes the following figure supplement(s) for figure 7:

**Figure supplement 1.** Characterizing neural activity following pC1-A activation.

---

that neural activity decays during a delay period following stimulation (*Figures 7–8*) and that behavioral effects of activation were also different following different delays. Specifically, we found that the highest levels of pC1 activation enhance receptivity but have an opposing effect on responses to male song (speeding females up instead of slowing them down). Slightly lower levels of pC1 activation (following a delay) bias females toward aggression and male-like behaviors. These effects on behavior may not naturally co-occur, but our optogenetic activation paradigm uncovers the scalable relationship between activation of different pC1 subtypes, their individual levels of activity, and distinct behavioral programs.

Our study also provides new insight into the neural mechanisms that contribute to changes in state on timescales of minutes (*Figures 7–8*). We used pan-neuronal imaging with registration to map responses that continue following pC1 optogenetic activation (previously, this technique had only been used to map sensory activity [*Pacheco et al., 2021*] and spontaneous activity [*Mann et al., 2017*]). We found that activation of pC1d/e neurons drives robust persistent neural activity throughout the posterior dorsal regions of the central brain (known to contain the processes of sexually dimorphic neurons [*Cachero et al., 2010*; *Kimura et al., 2015*], and overlapping with the Fru+ aIP-g neurons we identified as reciprocally connected with pC1d/e), lasting for minutes following activation. This is consistent with our behavioral observations - females still show elevated shoving and chasing even following a 6-min delay between optogenetic activation and the introduction of a male fly. Importantly, whether or not pC1 neurons themselves carry persistent neural activity has been debated (*Inagaki et al., 2014*; *Jung et al., 2020*; *Zhang et al., 2018*). Here, we find that in females, Dsx+ pC1 neurons, as well as multiple Fru+ neurons, including putative pC1 and aIPg cells, do indeed carry persistent neural activity in response to our activation protocol (*Figure 8*).

## pC1 neurons drive both aggression and receptivity in *Drosophila* females

We used unsupervised methods to identify the most prominent behaviors (beyond receptivity and responses to courtship song) produced *following* activation of pC1 neurons in virgin females - these include behaviors that resemble male courtship (female chasing the male) and aggression (female shoving the male) (*Figure 2*). Both behaviors are not typically observed in mature virgin females interacting with a male; this suggests that sensory cues from the virgin male do not inhibit these aberrant behaviors, but rather may enhance the persistent effects of pC1 activation (*Figure 2F*), most likely via visual inputs to aIPg neurons (*Schretter et al., 2020*). pC1 neurons also drive aggressive behaviors toward

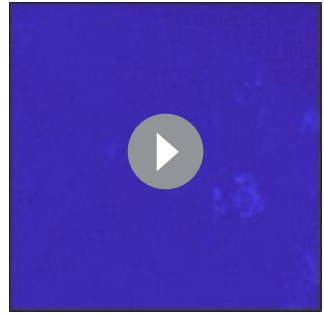

**Video 7.** Maximum z-projection (60 μm in Z) of the calcium response in a female expressing GCaMP6s pan-neuronally. Calcium response ((F(t) -Fo/Fo), color coded) is shown 5 min before, 5 min during and 9.5 min after pC1-A activation (using csChrimson). The movie is sped up 20 times.
https://elifesciences.org/articles/59502#video7

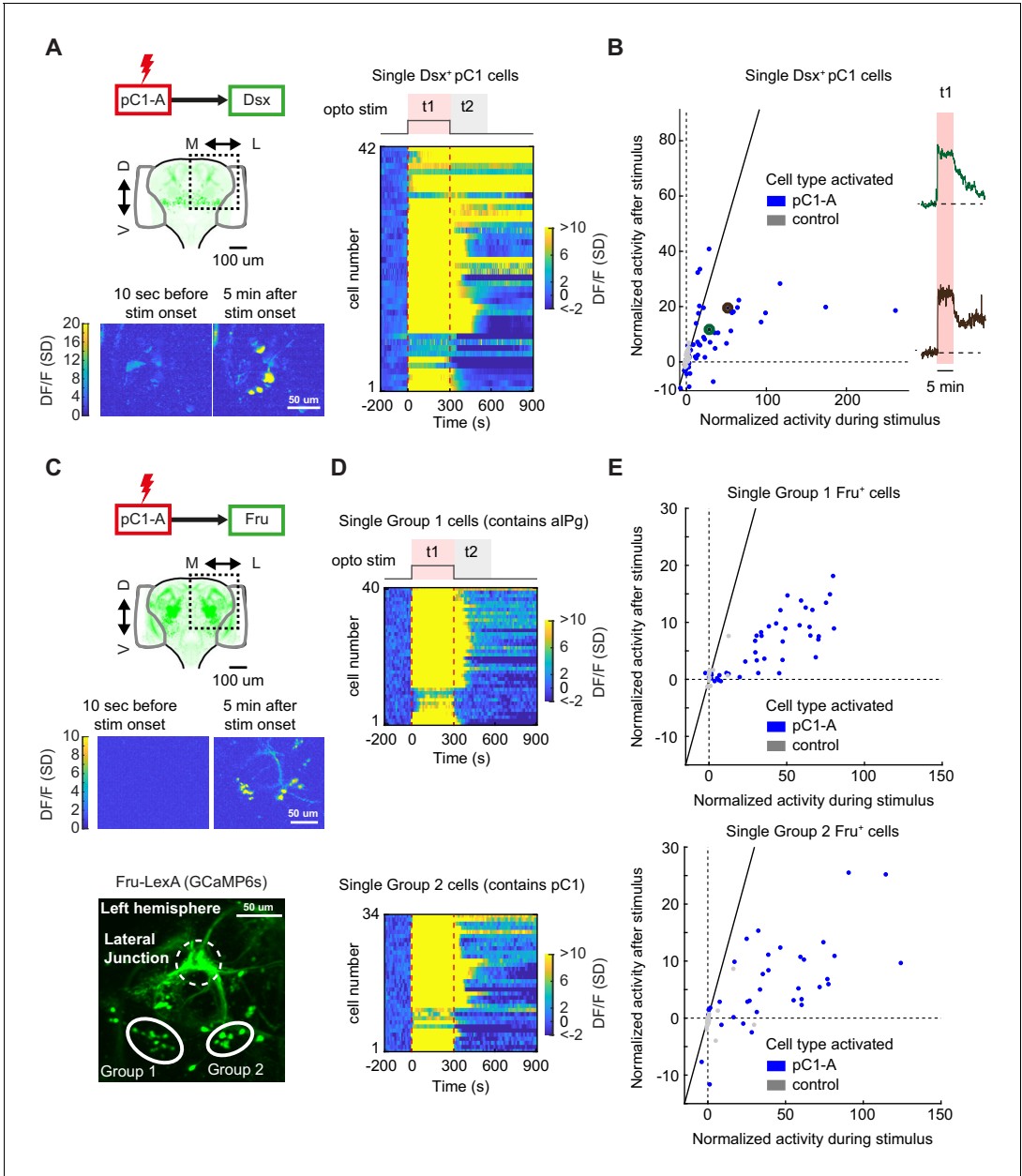

**Figure 8.** pC1d/e neurons drive persistent neural activity in Dsx+ and Fru+ cells. (**A**) Left: Activation of neurons in the pC1-A line and imaging of neural activity in all Dsx+ cell bodies. Activity in Dsx+ pC1 somas is shown 10 s before stimulus onset and 5 min after stimulus onset for an example experiment. Right: Normalized activity of Dsx+ pC1 somas ((F(t) - $F_0$)/$\sigma_o$; $F_0$ and F(t) are mean Fluorescence during baseline and fluorescence over time, respectively). Chrimson and TdTomato are expressed in pC1-A cells. GCaMP6s is expressed in Dsx+ cells. pC1 somas imaged following pC1-A activation (along with controls) are both shown if Ft1 > 3$\sigma_o$ ($\sigma_o$ - standard deviation during baseline, Ft1 is the mean fluorescence during t1) (see (**B**) for full set of imaged pC1 neurons). (**B**) Left: Mean Calcium response in Dsx+ pC1 somas during t1 (x-axis) versus during t2 (y-axis) for pC1-A activation and control. Normalized activity is defined as (F - $F_0$)/$\sigma_o$, where $F_o$ is the mean activity during baseline, $\sigma_o$ is the standard deviation during baseline, and F is the mean activity during t1 for x and t2 for y. Each dot represents a single segmented soma. Dots above the dashed line represent persistent responses following activation. All imaged pC1 neurons are shown (n = 8 flies, 58 ROIs for pC1-A, n = 5 flies, 25 ROIs for controls). Right: Example traces of (F - $F_o$)/$F_o$ from two individual pC1 cells, showing different response decays after stimulus offset (matching the results from **Figure 7D**). Corresponding points are enlarged and marked in green and purple in B (left). (**C**) Top: Activation of neurons in the pC1-A line and imaging of neural activity in Fru+ cell bodies. Bottom: Spatial pattern of Fru+ cell bodies imaged (here, a left hemibrain is shown (Z-projection of Fru+ neurons expressing GCaMP6s)). Fru+ cell body groups 1 and 2 are defined based on their spatial location - group 1 likely contains Fru+ pC2 and aIPg neuronal cell bodies, whereas group 2 likely contains Fru+ pC1 cell bodies (**Figure 8—figure supplement 2**). (**D**) Normalized activity of Fru+ somas from group 1 (Top) or group 2 (Bottom). (**E**) Mean activity in Fru+ cell bodies from group 1 (left, n = 9 flies, 46 ROIs for pC1-A, n = 5 flies, 13 ROIs for control) and group 2 (right, n = 9 flies, 37 ROIs for pC1-A, n = 5 flies, 24 ROIs for control) during t1 (x-axis) versus during t2 (y-axis) following pC1-A activation and in controls (see Key

*Figure 8 continued on next page*

*Figure 8 continued*

Resources Table for full genotypes). Data were analyzed and plotted as in (**B**). Dots above the dashed line represent persistent responses following activation.

The online version of this article includes the following figure supplement(s) for figure 8:

**Figure supplement 1.** Responses in Dsx+ pC2, pCd1 and pCd2 cells following pC1-A stimulation.
**Figure supplement 2.** Delineating Fru+ cells in Group 1.

females during stimulation (*Palavicino-Maggio et al., 2019*; *Schretter et al., 2020*), but whether the quality of aggression generated toward males versus females is similar remains to be determined. As one of our manually scored behaviors, 'female approaching' (*Figure 2—figure supplement 3A*), begins from a distance greater than four body lengths from the male fly (a distance at which it may be difficult to discern male from female [*Borst, 2009*]) and often ends with shoving or circling (see *Video 3*), we hypothesize that pC1 activation most likely drives persistent behaviors toward another fly, and not specifically a male or female fly, consistent with (*Schretter et al., 2020*).

What is the role of female aggression? Female aggression, whether toward males or females, has been previously reported across model systems (*Huhman et al., 2003*; *Stockley and Bro-Jørgensen, 2011*; *Woodley et al., 2000*). In *Drosophila*, female-female fights over food source are strongly stimulated by the receipt of sperm at mating (*Bath et al., 2017*), and include both patterns that are common with male aggression (such as shoving and fencing) and female-only patterns (*Nilsen et al., 2004*). Female-male aggression was reported in the context of rejecting behavior in mated, immature, or older females (*Cook and Connolly, 1973*). The behavioral changes in our study do not mimic those in a mated female, as we also observe that pC1 activation drives enhanced receptivity. Although we have not confirmed which pC1 cell types control receptivity, our work reveals a separation: pC1d/e neurons are sufficient to drive persistent shoving/chasing, but do not have a persistent effect on female receptivity (*Figure 4A–C*), while separate pC1 neurons that control receptivity modulate the pathways that control chasing and aggression (*Figure 4G–I*). Recent work reveals that pC1a neurons are modulated by the sex peptide receptor pathway, such that following mating (when sex peptide is transferred), pC1a neurons are inhibited (*Wang et al., 2020b*). Because we find that pC1a provides direct input to pC1d and pC1e neurons (*Figure 6A*, Table 1; see also *Schretter et al., 2020*), we speculate that pC1a provides this receptivity information. Interestingly, work in male flies suggests a separation in pC1 subsets that control courtship versus aggression (*Koganezawa et al., 2016*), with reciprocal inhibitory influences between persistent courtship and aggression, following pC1 activation (*Hoopfer et al., 2015*). Although the phenotypes are sex-specific (male singing vs female receptivity; male tussling vs female shoving), and the pC1 subsets driving these behaviors are sex-specific (P1 in males, pC1d/e in females), this suggests some common architecture. Ultimately, comparing the connectomes of male and female brains, combined with functional studies, should elucidate both similarities and differences.

Because courtship interactions unfold over many minutes (*Coen et al., 2014*), we postulate that the changes in brain state we observed following pC1d/e activation may occur naturally as females receive continual drive to pC1 neurons. Our connectomic analyses reveal inputs to pC1d/e neurons from AVLP cells (*Figure 6*; *Figure 6—figure supplement 1*). The AVLP contains multiple auditory cells (*Baker et al., 2020*) - this is consistent with our patch clamp recordings of pC1d/e neurons (*Figure 4—figure supplement 1*), showing auditory activity, and also with prior work on auditory responses in female, but not in male pC1 neurons (*Deutsch et al., 2019*). Thus, during natural courtship interactions, male song should drive pC1d/e neurons – in combination

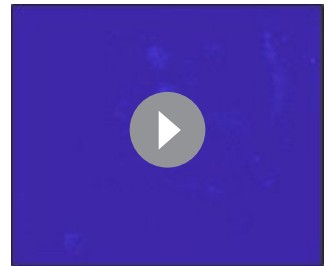

**Video 8.** Maximum Z-projection of the calcium response in a female expressing GCaMP6s in Dsx+ cells. pC1 cells in the left hemisphere are shown. Calcium level ((F(t) -Fo/Fo), color coded) is shown 5 min before, 5 min during and 9.5 min after pC1-A activation (using csChrimson). The movie is sped up 20 times.
https://elifesciences.org/articles/59502#video8

with other inputs, this may shift behaviors from receptive ones toward chasing and shoving.

## Recurrent circuitry and persistent neural activity

FlyWire enabled a systematic search for all the synaptic partners of a single pC1d and pC1e cell. Using manual and automatic synapse detection (*Buhmann et al., 2019*), we found that pC1d is reciprocally connected with aIPg-b and aIPg-c cells and that aIPg-b and aIPg-c cells are also interconnected – pC1e also shows reciprocal connectivity with aIPg-b neurons. We also identified pC1d and aIPg-c cells in a separate EM volume of an adult female brain, the hemibrain (*Scheffer et al., 2020*), and found similar results (see *Schretter et al., 2020* for a thorough analysis of the pC1d connectome in the hemibrain). Because FlyWire is based on the FAFB dataset of the entire adult female brain (*Zheng et al., 2018*), our search for synaptic partners of pC1d and pC1e could completely cover both hemispheres, showing that, for example, pC1d is reciprocally connected to itself in the contralateral hemisphere, and is presynaptic to some contralateral aIPg cells. Comparisons between connectivity diagrams in the two female EM volumes of FAFB (segmented in FlyWire) and the hemibrain will continue to be of value for future studies.

Synaptically recurrent neural networks have been proposed to contribute to persistent neural activity lasting for seconds, and to underlie processes like accumulation of evidence and working memory (*Aksay et al., 2007*; *Barak and Tsodyks, 2007*; *Major and Tank, 2004*; *Mante et al., 2013*; *Seung, 1996*; *Wang, 2008*; *Zylberberg and Strowbridge, 2017*). For example, in *Drosophila*, recurrent excitatory loops in the ellipsoid body of the central complex contribute to stabilization of a heading navigation signal over timescales of seconds (*Turner-Evans et al., 2020*). Internal states underlying social behaviors, however, persist on much longer timescales of minutes to hours (*Chen and Hong, 2018*). Neuromodulation, via hormones or peptides, is thought to support longer timescales of persistence (*Adolphs and Anderson, 2018*; *Bargmann, 2012*; *Marder, 2012*; *Zelikowsky et al., 2018*). Our work now links a strongly recurrent neural network (pC1d/e to aIPg) to both minutes-long persistent neural activity and minutes-long changes in behavior. Work from *Schretter et al., 2020* demonstrates that pC1d, pC1e, and aIPg neurons are all cholinergic, suggesting these neurons make up an excitatory neural network. While our study found persistent changes in behavior following pC1d/e activation, the *Schretter et al., 2020* study did not – this discrepancy could arise from a number of methodological reasons, including differences in the optogenetic activator, timescale of activation, or in behavioral protocols. Nonetheless, *Schretter et al., 2020* demonstrate that aIPg-b activation drives persistent female aggression following activation, and that increasing stimulation of aIPg-b, increases the amount of persistent aggression. This is consistent with a model in which strong activation of pC1d/e neurons recruits aIPg-b neurons (via their excitatory interconnections) to drive, at least, persistent shoving.

Work in male flies has proposed a recurrent circuit motif including pCd neurons that contributes to persistent aggressive behaviors (*Jung et al., 2020*) or even longer term changes in mating drive (driven by male-specific P1 neurons, but without mapping the synaptic connections). Our studies of pCd, revealed that, in females, it does not contribute to the persistent arousal state (*Figure 2—figure supplement 2*), nor is it persistently active following pC1d/e activation (*Figure 8—figure supplement 1*). In males, activating P1 neurons for 5 s was enough to induce some persistent activity in pCd cells (*Jung et al., 2020*). We did not measure neural responses to pC1d/e stimulation shorter than 2 min, and so whether shorter activation periods drive some persistent neural activity in females remains open. While pC1d/e neurons are female-specific, aIPg neurons are present in both males and females – it will be interesting to determine what role the aIPg neurons play in male brains and whether they represent a shared component of persistence in the two sexes.

Finally, it is important to point out that recent EM connectomic work in *Drosophila* using FAFB (via manual tracing) has revealed a number of recurrent or reciprocal circuits throughout the brain (*Dolan et al., 2018*, *Sayin et al., 2019*; *Turner-Evans et al., 2020*). Half of the outputs of pC1d are the aIPg neurons, and many of these neurons (several aIPg-b and aIPg-c cells) are strong inputs. Our identification of a recurrent circuit motif containing pC1d links it to the persistent neural activity and changes in behavior we observe following activation, especially in Fru+ somas. This is an important first step toward determining how recurrent neural networks contribute to such long timescales of persistence.

# Materials and methods

**Key resources table**

| Reagent type (species) or resource | Designation | Source or reference | Identifiers | Additional information |
|---|---|---|---|---|
| Genetic reagent (*D. melanogaster*) | NM91 | *Coen et al., 2014* | | A gift from Peter Andolfatto |
| Genetic reagent (*D. melanogaster*) | Dsx-Gal4 (III) | *Rideout et al., 2010* | | A gift from Stephen Goodwin |
| Genetic reagent (*D. melanogaster*) | Dsx-LexA::P65 (III) | *Zhou et al., 2015* | | A gift from Bruce Baker |
| Genetic reagent (*D. melanogaster*) | UAS-2xEGFP (II) | | BDSC: 6874; RRID:BDSC_6874 | |
| Genetic reagent (*D. melanogaster*) | R71G01-LexA (II) | *Pan et al., 2012* | BDSC: 54733; RRID:BDSC_54733 | |
| Genetic reagent (*D. melanogaster*) | R41A01-LexA (II) | *Zhou et al., 2014* | BDSC: 54787; RRID:BDSC_54787 | |
| Genetic reagent (*D. melanogaster*) | 8xLexAop2-FLP (II) | | BDSC: 55820; RRID:BDSC_55820 | |
| Genetic reagent (*D. melanogaster*) | UAS > STOP > TNT (III) | *Stockinger et al., 2005* | | A gift from Barry Dickson |
| Genetic reagent (*D. melanogaster*) | UAS-ReachR (III) | *Inagaki et al., 2014* | BDSC: 53749; RRID:BDSC_53749 | |
| Genetic reagent (*D. melanogaster*) | VT25602.p65ADZp (II) | *Wu et al., 2019* | | A gift from Barry Dickson |
| Genetic reagent (*D. melanogaster*) | VT2064.ZpGAL4DBD (III) | *Wu et al., 2019* | | A gift from Barry Dickson |
| Genetic reagent (*D. melanogaster*) | UAS(FRT.mCherry)ReachR (III) | *Inagaki et al., 2014* | BDSC: 53740; RRID:BDSC_53740 | A gift from David Anderson |
| Genetic reagent (*D. melanogaster*) | R71G01-p65.AD (II) | *Dionne et al., 2018* | BDSC: 70798; RRID:BDSC_70798 | A gift from Gerry Rubin |
| Genetic reagent (*D. melanogaster*) | Dsx.DBD (III) | *Pavlou et al., 2016* | | A gift from Stephen Goodwin |
| Genetic reagent (*D. melanogaster*) | 10xUAS-IVS-myr::GFP (II) | | BDSC: 32198; RRID:BDSC_32198 | |
| Genetic reagent (*D. melanogaster*) | W+,NorpA (X) | | BDSC: 9048; RRID:BDSC_9048 | |
| Genetic reagent (*D. melanogaster*) | VT25602-Gal4 (III) | | | A gift from Barry Dickson |
| Genetic reagent (*D. melanogaster*) | R57C10-LexA (II) | | BDSC: 52817; RRID:BDSC_52817 | A gift from Gerry Rubin |
| Genetic reagent (*D. melanogaster*) | 8xLexAop-mCD8tdTomato (III) | | | A gift from Yuh Nung Jan |
| Genetic reagent (*D. melanogaster*) | 13xLexAop-IVS-GCaMP6s (III) | | BDSC: 44274; RRID:BDSC_44274 | |
| Genetic reagent (*D. melanogaster*) | 10xUAS-IVS-Syn21-Chrimson-tdT-3.1 (X) | *Hoopfer et al., 2015* | | A gift from Allan Wong |
| Genetic reagent (*D. melanogaster*) | 13xLexAop2-IVS-Syn21-opGCaMP6s (X) | | | A gift from Allan Wong |
| Genetic reagent (*D. melanogaster*) | 20xUAS-IVS-CsChrimson.mVenus (X) | | BDSC: 55134 RRID:BDSC_55134 | A gift from Vivek Jayaraman |
| Genetic reagent (*D. melanogaster*) | Fru[P1.LexA] (III) | | BDSC: 66698 RRID:BDSC_66698 | A gift from Bruce Baker |
| Antibody | Mouse anti-brp | mAb DSHS - developmental studies Hybridoma bank | #nc82; RRID:AB_2314866 | (1:20 dilution) |

*Continued on next page*

*Continued*

| Reagent type (species) or resource | Designation | Source or reference | Identifiers | Additional information |
|---|---|---|---|---|
| Antibody | Chicken anti-GFP (Polyclonal) | Invitrogen | #A10262, RRID:AB_2534023 | (1:2000 dilution) |
| Antibody | Rabbit anti-dsRed (Polyclonal) | Takara Bio | #632496, RRID:AB_10013483 | (1:500 dilution) |
| Antibody | Goat anti-mouse Alexa-Flour 568 (Polyclonal) | Invitrogen | #A-11004, RRID:AB_2534072 | (1:400 dilution) |
| Antibody | Goat anti-chicken Alexa-Flour 488 (Polyclonal) | Invitrogen | #A11039, RRID:AB_142924 | (1:300 dilution) |
| Antibody | Goat anti-rabbit Alexa-Flour 633 (Polyclonal) | Invitrogen | #A-21070, RRID:AB_2535731 | (1:400 dilution) |
| Chemical compound, drug | All trans-Retinal | Sigma-Aldrich | #R2500 | |
| Software, algorithm | MATLAB | Mathworks | RRID:SCR_001622 | |
| Software, algorithm | SLEAP | https://sleap.ai/ | | |
| Software, algorithm | Illustrator | Adobe | RRID:SCR_010279 | |

## Fly genotype table

| Figure panel | Genotype | Additional information |
|---|---|---|
| 1A | UAS-2xEGFP; Dsx-Gal4 | Immunostaining |
| 1C-D | w;R71G01-LexA/8xLexAop2-FLP; Dsx-Gal4/UAS > STOP > TNT | Express TNT in pC1-Int |
| 1S1B | w;R41A01-LexA/8xLexAop2-FLP; Dsx-Gal4/UAS > STOP > TNT | Express TNT in pCd1 neurons |
| 1C-D, 1S1B | w;+/8xLexAop2-FLP;Dsx-Gal4/UAS > STOP > TNT | Control for TNT expression in pC1-Int and in pCd1 |
| 1 F-G, 2, 4 F-H, 1S1D, 2S1, 2S2B-C, 2S3 | w;R71G01-LexA/8xLexAop2-FLP; Dsx-Gal4/10xUAS > STOP > ReachR | Express ReachR in pC1-Int neurons; ATR+ experimental, ATR- control |
| 1S1C, 2S2A | w;R41A01-LexA/8xLexAop2-FLP; Dsx-Gal4/10xUAS > STOP > ReachR | Express ReachR in pCd1 neurons; ATR+ experimental, ATR- control |
| 4D-E | w;R71G01.AD/+;DSX.DBD/UAS-ReachR | Express ReachR in 'pC1-s' neurons (behavioral experiment); ATR+ for experimental, ATR- for control |
| 4A-C, F, J | w;VT25602.AD/+;VT2064.DBD/10xUAS-ReachR | Express ReachR in 'pC1-A' neurons (behavioral experiment); ATR+ for experimental, ATR- for controls. pC1-A is similar to pC1dSS3 in *Schretter et al., 2020*. |
| 4S1 | w;UAS-2xEGFP/+;VT25602-Gal4/+ | Express GFP in pC1d/e neurons for patch clamp recordings |
| 3D (top), 3S2C | w;R71G01.AD/UAS-10x-IVS-myr::GFP;Dsx.DBD/+ | Immunostaining (pC1-S) |
| 3D (bottom), 3S2B | w;VT25602.AD/UAS-10x-IVS-myr::GFP;VT2064.DBD/+ | Immunostaining (pC1-A) |
| 3S2A | R71G01-LexA/8xLexAop2-FLP; Dsx-Gal4/UAS > STOP > myrGFP | Immunostaining (pC1-Int) |
| 7B-E, 7S1A | w+NorpA[36],20xUAS- csChrimson.mVenus; R71G01.AD/R57C10-LexA; Dsx.DBD/8xLexAop-mCD8tdTomato, 13xLexAop-GCaMP6s | pC1-S activation, measure pan-neuronal Ca response; ATR+ |
| 7B-J, 7S1A-B | w+NorpA[36],20xUAS-csChrimson.mVenus; VT25602.AD/R57C10-LexA; VT2064.DBD/8xLexAop-mCD8tdTomato, 13xLexAop-GCaMP6s | pC1-A activation, measure pan-neuronal Ca response; ATR+ |

*Continued*

| Figure panel | Genotype | Additional information |
|---|---|---|
| 7B-E,I-J, 7S1A | w+NorpA[36],20xUAS-csChrimson.mVenus; R57C10-LexA/CyO;8xLexAop-mCD8tdTomato, 13xLexAop-GCaMP6s/TM6B,tb | Control; ATR+ |
| 8A-B, 8S1 | 10xUAS-Chrimson.tdTomato, 13LexAop2-GCaMP6S/W+NorpA[36], 20xUAS-csChrimson.mVenus; VT25602.AD/CyO;Dsx-LexA/TV2064.DBD | pC1-A activation, measure Ca response in Dsx+ neurons; ATR+ |
| 8A-B, 8S1 | 10xUAS-Chrimson.tdTomato,13LexAop2-GCaMP6s/+; Sp/CyO;Dsx-LexA/TM6B,tb | Control; ATR+ |
| 8C-E, 8S1 | 10xUAS-Chrimson.tdTomato,13LexAop-GCaMP6s/+; VT25602.AD/CyO; Fru-LexA/VT2064.DBD | pC1-A activation, measure Ca response in fru+ neurons; ATR+ |
| 8C-E | 10xUAS-Chrimson.tdTomato,13LexAop-GCaMP6s; Sp/CyO;Fru-LexA/Tm6Tb | Control; ATR+ |

## Fly stocks

All flies were raised at 25°C on standard medium in a 12 hr light/12 hr dark cycle at 60% relative humidity. Female flies used for optogenetic experiments were fed with food that contained all-trans-retinal (Sigma R2500-100MG; ATR concentration is 1 mM) for a minimum of 3 days post-eclosion. Control flies were raised on regular fly food after eclosion. Both experimental and control female flies used for optogenetic experiments, were reared post-eclosion in dark blue acrylic boxes (acrylic available from McMaster-Carr, #8505K92).

The full details on genotypes for the flies used in this study and their source are in Key resources table and in Fly genotype table.

## Behavioral experiments

All behavioral experiments were carried out using a behavioral chamber (diameter ~25 mm) tiled with 16 microphones (NR23158, Knowles Electronics; *Figure 1B* and *Videos 1–3*), and connected to a custom amplifier (*Arthur et al., 2013*). Audio signals were recorded at 10 KHz, and the fly song was segmented as previously described (*Arthur et al., 2013*; *Coen et al., 2014*). A point gray camera (FL3-U3-13Y3M; 1280 × 960) was used to record fly behavior from a top view (see *Figure 1B*, S1A) at 60 frames per second using custom written software in Python and saved as compressed videos (H.264). Virgin females (see Fly genotype table for genotype used in each experiment) and wild-type NM91 virgin males (both males and females were 3–7 days old) were used for all behavioral experiments. For inactivation experiments (*Figure 1C–D*, *Figure 1—figure supplement 1B*), a male and a female were introduced into the behavioral chamber simultaneously. For activation experiments, a female was positioned in the behavioral chamber, and red light (a ring of 6 LEDs, 627 nm, LuxeonStar) was then delivered at 1.1 mw/mm$^2$ (±5% across the chamber) at 100 Hz (50% duty cycle) for 5 min (*Figure 1E*) or for 2 min (*Figure 4J*). Following stimulus offset, a male was introduced with either no delay (d0), or after a 3 or 6 min delay (d3, d6). We collected data from the time the male was introduced (t = 0) until 30 min or when the flies copulated, whichever came first. We chose to use ReachR (*Inagaki et al., 2014*) for activation experiments rather than using the more sensitive red shifted channelrhodopsin csChrimson (*Klapoetke et al., 2014*) to minimize optogenetic activation due to background light. The percent of flies that copulated as a function of time (*Figure 1C,F*, *Figure 1—figure supplement 1B–C*) was calculated from the time the male was introduced (t = 0) and until t = 30 min.

## Tracking centroids and headings

Each video frame was analyzed by first finding the fly centroid, then detecting the location of the thorax and head which formed the heading vector for each fly. Having microphones at the chamber bottom results in a highly inhomogeneous background (*Figure 1—figure supplement 1A*) posing a major challenge for accurate tracking.

The centroid was found in one of two ways that yielded similar results. In the first method, the inhomogeneous background of the video was found by taking the median across all frames. Because the animals move throughout the video, finding the median pixel usually does not contain any pixels containing an animal's body. However, as animals occasionally sit for long periods, they can become part of the background. To avoid this, we divided the video into 10 shorter videos of equal length and found the median 'frame' (median set of pixel values) for each sub-video. We then created a median frame by computing the median across these medians. Each video frame then had this background subtracted to identify pixels that were potentially part of each fly. These pixels were smoothed by a series of operations using the OpenCV Python package and then thresholded. Using OpenCV, we identified all contours surrounding collections of pixels and any smaller or larger than some predefined threshold (less than half the size of a typical 'fly' or more than twice its size) were discarded. The remaining pixels were then clustered via k-means. The number of clusters were iteratively increased until the compactness of each cluster reached some threshold. The least-compact clusters were discarded, and the remaining pixels were clustered again with k-means with k = 2 to identify the two clusters corresponding to the animals. These clusters were then fit with an ellipse to identify the centroid of each animal. In the second method, we trained a deep convolutional network to detect all instances of individual body parts (head, thorax) within each frame using a modified version of LEAP (*Pereira et al., 2019*, or SLEAP [*Pereira et al., 2020*] https://sleap.ai/; 544 labeled frames were used for training; *Figure 1—figure supplement 1Ai-ii* and *Video 1*). Using the same software and neural network architecture, a separate network was then trained to group these detections together with the correct animals by inferring part affinity fields (*Figure 1—figure supplement 1Aiii*; *Cao et al., 2017*). This enabled estimation of the vector that represents fly heading for both flies (*Figure 1—figure supplement 1Aiv*).

## Linear classification of single frames

Seventeen parameters were extracted for each frame based on the tracking of male and female centroid and heading (*Figure 2A*), describing either the female movements (fFV – female forward velocity; fLS – female lateral speed; fFA/fLA – female forward/lateral acceleration; fRS – female rotational speed), male movements (similar to female movements – mFV/mLS/mFA/mLA/mRS), or male-female interaction (mfDist – male-female distance; fmAngle – female heading relative to the female-male axis, mfAngle – male heading relative to male-female axis; fmFV or fmLS – female speed in the male direction or in the perpendicular axis; mfFV or mfLS – male speed in the female direction or perpendicular). Using 17 parameters for each frame, we trained binary support vector machine (SVM) linear classifiers to find the parameters (dimensions) that best separate the groups. We first trained classifiers that separate between frames that belong to experimental flies (class 1, pC1-Int activated, either one condition - d0/d3/d6 or all groups together, d0-d6), and controls (class 2). We trained 90 classifiers, randomly choosing a set of 3000 frames from each class ('training set'; non-overlapping - the same frame was never used in two classifiers; increasing the number of frames beyond 3000 did not increase performance). We used the MATLAB R2019b procedure *fitcsvm* (MathWorks, Natick, MA), with a linear kernel. We then used a separate set of 30,000 frames per class for each classifier ('validation set; the same frame was never used twice, either between classifiers or between sets) to test the performance of each classifier (fraction of frames correctly classified). We then choose the 30 best-performing classifiers (*Figure 2B* for control vs d0-d6). We used a third set of frames for each classifier (30,000 frames/class, again – with no overlap with other sets) to measure the performance of each classifier. The MATLAB function *predict* was used to find the SVM-predicted class for each frame in the validation or train set. Performance was calculated as the percent of frames correctly classified (*Figure 2C*). For each weight (out of the 17; control vs d0-d6), we looked at the distribution coming from the 30 independent classifiers, and tested whether the mean was significantly different than zero (*Figure 2—figure supplement 1A*).

We used a two-sample t-test to measure the probability that the mean weight associated with each parameter is different from zero (*Figure 2—figure supplement 1A*). We found 8 out of the 17 parameters to be highly significant (*p<0.0001).

## Clustering behaviors based on single frames

The eight most significant parameters found by the SVM classifier (see previous section) were used for classification. We took the same number of frames from each group (control/d0/d3/d6) - 357,997 frames (99.4 min) per group, corresponding to the number of frames in the smallest group (d6). We repeated the clustering 30 times (*Figure 2D*, black dots), each time selecting 99.4 min of data from each one of the other groups (d0, d3, d6, control) randomly (with replacements – the same frame could be used in multiple repeats), therefore having >1.4 million frames for clustering on each repeat. The sets are not independent (overlapping frames between repeats) and no statistical test was performed over the repeats. After z-scoring each parameter (over all the frames in a given repeat), k-means clustering was performed (using MATLAB function *kmeans*), allowing 20 clusters and a maximum of 500 iterations (other parameters set to default). We found that the first seven largest clusters (cluster size being the number of frames in the cluster) capture 90.4% of the frames, averaged over repeats (*Figure 2D*, inset). To match clusters between repeats (for each cluster number in repeat 1, find the corresponding cluster number in repeats 2–30), we used the smallest distance between clusters, by calculating the mean square error over the weights (the variability in weight size across repeats is shown in *Figure 2—figure supplement 1C*).

## Machine-learning-based classification of behavioral epochs

The Janelia Automatic Animal Behavior Annotator (JAABA; http://jaaba.sourceforge.net/; *Kabra et al., 2013*) was used to detect epochs of 'female shoving' and 'female chasing'. Two independent classifiers ('shoving classifier', 'chasing classifier') were trained, one for each behavior. We used the automatic segmentations to find examples for shoving and chasing epochs, used as a first step in training each classifier. We then added example epochs (positive and negative examples are used for each classifier), in an iterative manner (using examples where the classifiers made wrong predictions). Altogether we used 24,222 frames (6.7 min) to train the 'shoving classifier', and 11,941 frames (3.3 min) for the 'chasing classifier'.

The classifiers were based on the 17 parameters defined above (denoted as 'per-frame' features), as well as on 'window features' ('mean', 'min', 'max', 'change', 'std', 'diff_neighbor_mean', 'diff_neighbor_min', 'diff_neighbor_max', 'zscore_neighbors' with a window radius of 10 and default 'windows parameters'), therefore taking into account longer timescales for classification, rather than the single frames we used for SVM classification and k-means clustering (see *Figure 2—figure supplement 1C* for comparison). We cross validated each classifier before applying the classification on all the data, using the cross-validation procedure available in JAABA package (with default parameters). A total of 94.2% of the frames annotated by the user as shoving were correctly classified as shoving, while 92.8% of the frames annotated as no-shoving were classified as no-shoving. For the 'chasing classifier', we got 96% and 90.8% success in classifying chasing and no-chasing. The trained shoving classifier was used to annotate each frame as belonging or not belonging to 'female shoving' epoch, and the trained chasing classifier was used independently to classify each frame as belonging or not-belonging to a 'female chasing' epoch. The same classifiers were used for all experimental conditions (5 min or 2 min activation duration, d0/d3/d6 delay conditions).

## Manual tracking annotation of behaviors

We annotated a subset of the data manually (pC1-Int, d0-d6), to confirm our automatic behavior detection, as well as in search for more rare events, or events that are not captured due to tracking issues. Three behaviors were annotated by two observers: female shoving, female chasing and circling (*Figure 2—figure supplement 3A–C*). The two observers annotated different sets of movies, while a small subset (n = 5 movies) were annotated by the two observers and we confirmed that both detected three behaviors (shoving, chasing, circling) similarly. Female circling was not detected by our automated procedures for two reasons. First, during circling male and female bodies often overlap, causing large errors in heading detection. Second, these events are relatively sparse. One observer also detected three other rare behaviors: head butting, female mounting (*Figure 2—figure supplement 3C*, *Video 3*) and wing extension (*Figure 2—figure supplement 3D–E*, *Video 2*).

## Statistical analysis

Statistical analysis was performed using MATLAB (Mathworks, Natick, MA) procedures, and corrected for multiple comparisons using the Bonferroni correction when appropriate. The details on the statistical test used are listed under the Results section and the Figure legends. Black lines between two groups indicate a statistically significant difference between the groups after multiple comparison correction, while a red line indicates that the difference is statistically significant only when multiple comparisons test is not used. To test for significant differences in copulation rate, we used Cox's proportional hazards regression model, using the MATLAB procedure *coxphfit*. 'Censoring' was used to account for the fact that some flies copulated within the 30 min time window (after which the experiment was terminated), while others did not. The correlation between female velocity and male song (*Figure 1D,G*) was done as previously described (*Clemens et al., 2015*). Briefly, female absolute speed and male song were averaged over 1 min windows. In each window, we calculated the mean value of female (absolute) speed, bout amount (the total amount of song in the window), bout number (the number of song bouts in the window), and bout duration (the mean bout duration in the window). Then, for each condition, we calculated the correlation between female speed and male song by pooling all windows for a given group together. The MATLAB procedure *corr* was used to calculate the Pearson correlation, and one way analysis of covariance (ANOCOVA) was used to compare the slopes (x,y being the male song and the female speed) between groups using aoctool (MATLAB). The 30 SVM (Support Vector Machine) classifiers (*Figure 2B–C*, *Figure 2—figure supplement 1C*) were trained using non-overlapping sets of frames and are therefore considered independent. One-sample t-test was used to calculate a test decision for the null hypothesis that the 30 weight values (for a given parameter) come from a normal distribution with a mean of zero (and unknown variance). For each parameter, -log(P) is shown, and a vertical dashed indicates $p<10^{-4}$ (*Figure 2—figure supplement 2C*).

## Immunostaining

Flies were dissected in S2 insect medium (Sigma #S0146). Dissected brains were moved through six wells (12ηl/well) containing a fixation solution 4% paraformaldehyde, Electron microscopy sciences #15713 in PBT (0.3% Triton in PBSX1; Triton X-100 Sigma Aldrich #X100; PBS - Cellgro #21–040), before sitting for 30 min on a rotator at room temperature. Following fixation, brains were moved through six wells containing PBT, 15 min in each well. Then, brains were transferred through four wells containing a blocking solution (5% Goat Serum in PBT; Life Technologies #16210–064), and sitting in the last well for 30 min. Brains were then moved to a solution containing primary antibodies (see below) and then incubated for two nights at 4°C (sealed and light protected). After eight washes (20 min per wash) in PBT, brains were incubated overnight with secondary antibodies. After eight washed (20 min each) in PBT, brains were placed on a slide (Fisher Scientific #12-550-15), between two zero numbered coverslips used as spacers (Fisher Scientific 12-540B) and under a coverslip (Fisher Scientific #12-542B), and Vectashield (Vector Laboratories) was applied. Nail polish was used to seal around the center coverslip edges, and brains were stored in dark at 4°C overnight to harden, before imaging. See Key Resources Table for the list of antibodies used in this study. Imaging was done using a Leica confocal microscope (TCS SP8 X). *Figure 1A* was modified from *Deutsch et al., 2019*.

## Identification and proofreading of neurons in FlyWire

Neurons in a complete EM volume of an adult female brain (*Zheng et al., 2018*) were automatically reconstructed in FlyWire (flywire.ai, [*Dorkenwald et al., 2020*]). Within FlyWire, we first searched for reconstructed segments that match the morphology of known pC1 cells. We used anatomical landmarks to find the bundle that projects dorsally from pC1 cells bodies (*Figure 1A*, red arrow). We then looked at two cross-sections of this bundle in each hemisphere (*Figure 3B* and *Figure 3—figure supplement 1*) and scanned systematically all the segments that pass through this bundle. Based on known morphology of female pC1 cells (*Deutsch et al., 2019*; *Kimura et al., 2015*; *Zhou et al., 2014*), we defined cells as pC1 when they crossed through the pC1 bundle, and also projected to the lateral junction (*Figure 1A*). Similarly, pC2l cells were found by looking through the pC2l bundle (see *Deutsch et al., 2019*; we refer to these cells as pC2 below). The aIPg cells were first found by searching for neurons synaptically connected to pC1d (see below), and other aIPg cells were found

by systematically exploring a cross section within the aIPg bundle (*Figure 6E* and *Figure 5—figure supplement 1*). pMN1 and pMN2 were found when mapping the pC1d synaptic partners, and then named pMN1 and pMN2 based on their morphology (*Deutsch et al., 2019*; *Kimura et al., 2015*). pC1 and aIPg cells were sorted manually into subtypes based on morphology (*Figure 3C* and *Figure 5—figure supplement 1B*).

Proofreading of a neuron was performed using the tools available in FlyWire (flywire.ai, [*Dorkenwald et al., 2020*]). In short, this process has two parts: (1) removing ('splitting') parts that do not belong to the cell ('mergers'), such as parts of glia or parts of other neurons (for example, when detecting two cell bodies in one segment), and (2) adding missing parts ('merging'). We had an average of 5.4 splits and 10.7 merges per neuron, and proofreading a single cell took 43 min on average (we measured the proofreading time for a subset of the cells we proofread). Proofreading was complete when no additional obvious mergers were found, and we couldn't identify missing parts at the edge of any processes. In some cases, the known morphology of the cell (e.g. pMN1 or pMN2) or the existence of other cells with similar morphology (in the same or the other hemisphere) were used to verify that no major processes were missing. Sorting cells into types was done manually, based on their morphology. pC1 was divided into five subtypes, and aIPg were divided into three subtypes.

Assigning names to known neurons we found in the EM volume was done solely based on morphology. It is possible, that in some cases (e.g. for pC1, pC2, or aIPg cells), some of the neurons we found are not actually Dsx+ or Fru+ cells. Additional work is needed to compare LM based and EM based morphologies, and to classify cell types based both on morphology and connectivity (*Scheffer et al., 2020*). Finding cell types in hemibrain version 1.1 with shared morphology was done for the major inputs and outputs of pC1d/e manually.

## Mapping synaptic inputs and outputs in FlyWire

We mapped all the direct inputs and outputs of a single pC1d neuron (*Figure 3C*) by manually detecting pre- and postsynaptic partners for this cell. After proofreading the cell (see details above), we looked systematically, branch by branch, for synaptic partners based on previously defined criteria. For a contact to be defined as a chemical synapse, it had to meet three conditions: (1) the presence of a synaptic cleft between the pre- and postsynaptic cells, (2) presynaptic active zone with vesicles near the contact point, and (3) one of two (or both) must exist: a presynaptic T-bar adjacent to the cleft (*Fouquet et al., 2009*) at the presynaptic terminal or a postsynaptic density (PSD, [*Ziff, 1997*]). In flies, PSDs are variable, and are often unclear or absent (*Prokop and Meinertzhagen, 2006*). Typically, we observed T-bars rather than PSDs, as a T-bar is easier to identify. Our criteria was slightly more conservative than the one used in *Zheng et al., 2018*, possibly leading to less false positives (wrongly assigned synapses), and more false negatives (missing synapses). On top of using a conservative criterion, synapses were missed for several other reasons, including missed detections by the manual observers (see comparison to automatic detection below) and signal to noise issues in the dataset. Once a synapse was detected, we then looked for the post-synaptic partner. Around 10% of the inputs to pC1d and about 60% of the outputs were short segments ('twigs'), that we could not connect to backbones in order to identify or proofread the connected neuron. The twigs were not restricted to a specific part of the pC1d cell, and we therefore believe that they do not impose a bias on the distribution of pC1d connections, though it is possible that specific output types (e.g. cells with thinner processes) are less likely to be detected.

Following the detection of pC1d synaptic partners, we mapped the inputs and outputs to pC1d in three steps. First, we manually proofread the input and output segments. Second, we eliminated cells that connect to pC1d with less than three synapses, to reduce the number of potential false positives, and to focus on stronger connections. We ended up having 78 input and 52 output cells. Third, we sorted cells manually into cell types based on morphology. Some cells were classified based on known morphology from light microscopy (pMN1, pMN2, pC1, pC2, aIPg). In order to look for connections between pC1 and pC2 cells (the largest sets of Dsx+ neurons) in an unbiased way (not focusing on specific types or individual pC1 or pC2), we first identified and proofread pC1 and pC2 cells. Synaptic connections between individual cells of pC1 or pC2 type were detected by manually inspecting the volume plane by plane. Once a pair of segments that came within proximity of one another was detected, we zoomed and looked for synaptic connections based on the criteria defined above.

We also used automatic detection of synapses in FAFB. FlyWire provides a mapping of the publicly available, automatically detected synapses from *Buhmann et al., 2019*.

We estimated the reliability of synapse detection by manually testing all the synapsis between pC1d (presynaptic) and two different output cells (a single aIPg-b and a single pC2-like). We found 61.7% (71/115) of the tested synapses to be true positives according to our criteria, 24.3% (28/115) redundant (two coordinates for the same synapse) and 13.9% false positives. By looking at pairwise distances between presynaptic coordinates, and using a minimum threshold of 150 nm, we were able to remove 27/28 of the redundant synapses, while eliminating only three true synapses, thus ending up having 85% (98/115) true positives. We therefore used this threshold whenever using the Buhmann detector. We detected all the inputs and outputs of a single pC1d (FlyWire coordinates 145817, 41367, 5139) and a single pC1e (FlyWire coordinates 142741, 44980, 4908) cells. Cells were included in the count only if they followed the following two criteria: (1) Connected to the inspected pC1d or pC1e with six synapses or more and (2) is part of a cell type (based on morphology, using manual classification) that includes at least one cell with strong connection (defined as 15 synapses or more) with pC1d or pC1e. Only synapses that belong to these cells were counted (see Table 1; *Figure 5F*; *Figure 6*; *Figure 6—figure supplement 1*).

Finding the best match in the single-clone dataset FlyCircuit for a given EM segment was done in two steps. First, an. swc file was generated for a given segment (using the automatically segmented cells rather than the proofread ones for technical reasons). Second, we performed an NBLAST search (*Costa et al., 2016*) either online (http://nblast.virtualflybrain.org:8080/NBLAST_on-the-fly/) or using 'natverse', an R package for neuroanatomical data analysis (*Bates et al., 2020*). For visualization purposes, we first created mesh files (.obj) for proofread neurons, and then used either Meshlab (http://www.meshlab.net/) to create images, or Blender (https://www.blender.org/) to create movies (see support/FAQ in https://flywire.ai/ for instructions on creating. swc and. obj files).

## In vivo calcium imaging

We imaged brain activity following pC1 optogenetic activation (though the microscope objective) under a two-photon custom made microscope (*Pacheco et al., 2021*) in females, using the calcium indicator GCaMP6s (*Chen et al., 2013*). Both GCaMP6s and the structural marker tdTomato (*Shaner et al., 2004*) were expressed pan-neuronally in blind flies (NorpA[36] mutant) using the nsyb enhancer (*Bussell et al., 2014*). For pC1 activation, we used the same temporal pattern as the one used in the behavioral experiments: 5 or 2 min of light on, at 100 Hz and 50% duty cycle. Imaging started 5 min before stimulus onset, where baseline activity was measured, and lasted 9.5 min after stimulus offset for whole-brain imaging and 30 min after stimulus offset for *doublesex* imaging. While the red shifted channelrhodopsin ReachR was used for behavior experiments to minimize optogenetic activation by background light, we used Chrimson for two-photon imaging for two reasons. First, to minimize the amount of bleed-through from the optogenetic activation light to the green photomultiplier tube (PMT). For this reason we also choose a longer wavelength of 700 nm (M700L4, Thorlabs, Newton, NJ with a band pass filter FF01-708/75-25, Semrock, Rochester, NY) that is well separated from the range of light that cross the green PMT entrance filter (Semrock FF01-593/40-25). Second, we wanted to minimize the amount of light needed for neuronal activation by using a more sensitive effector, to reduce the amount of heat accumulating in the brain during imaging (on top of the heating caused by the two-photon laser, whose power was limited to 15 mW). We are aware of possible differences in pC1 activation level between the behavioral and imaging experiments. Based on existing literature, we tried to choose an activation level for the imaging experiments that will roughly match the activation induced during behavior. In order to match the activation level between the behavioral experiments (ReachR, 627 nm light, intact fly) and the imaging experiments (csChrimson, 700 nm light, cuticle removed above the fly brain) we used data from existing literature. By comparing the amount of light needed for driving proboscis extension reflex (PER) in 100% of adult flies in *Inagaki et al., 2014* (ReachR, 1.1 mW/mm$^2$, 627 nm) to the level of light used to saturate PER score in *Klapoetke et al., 2014* (CsChrimson, 0.07 mW/mm$^2$, 720 nm), taking into account the different duty cycles used in the two studies and given the penetration rate through the cuticle (Based on *Inagaki et al., 2014*; *Figure 1A*, around 6% of the light penetrates at 627 nm), we choose a light intensity of 0.013 mW/mm$^2$.

A volume of ~307 $\times$ 307 $\times$ 200 µm$^3$ from the dorsal part of the central brain was scanned at 0.1 Hz (1.4 $\times$ 1.2 $\times$ 2 µm$^3$ voxel size), covering a complete dorsal quadrant (full anterior-posterior axis of

the central brain) which represents about 58.02 ± 3.97% of the whole hemisphere (mean ± SD, n = 28 animals). Volumetric data was processed as described in *Pacheco et al., 2021*. In brief, tdTomato signal was used to motion-correct volumetric time-series of GCaMP6s signal in XYZ axis (using the NoRMCorre algorithm [*Pnevmatikakis and Giovannucci, 2017*]). Volumes were spatially resampled to have isotropic XY voxel size of $1.2 \times 1.2 \times 2$ µm$^3$ (bilinear interpolation on X and Y axes), and temporally resampled to correct for different slice timing across planes of the same volume, and to align timestamps of volumes relative to the start of the optogenetic stimulation (linear interpolation). Next, the GCaMP6s signal was 3D-ROI segmented using the Constrained Nonnegative Matrix Factorization (CNMF) algorithm to obtain temporal traces and spatial footprints per segmented ROI as implemented in CaImAn (*Giovannucci et al., 2019*; *Pacheco et al., 2021*). In this algorithm, each ROI is defined as a contiguous set of pixels within the field of view that are correlated in time, these ROIs are initialized around locations with maximum variance across time. Spatially overlapping ROIs with correlated activity are merged (correlation coefficient >0.9); therefore, ROIs could have different sizes. Code to perform these processing steps are available at https://github.com/murthylab/FlyCaImAn (*Pacheco et al., 2021*). In addition, to further remove residual motion artifacts from the GCaMP6s signal, in particular slow drift over tens of minutes, we performed independent component analysis (ICA) on the tdtomato ($F_{tdtomato}$) and GCaMP6s ($F_{GCAMP}$) signal for each ROI independently, similar to *Scholz et al., 2018*. To remove opto-related artifact bleeding through the red channel, $F_{tdtomato}$ was linearly interpolated from 20 s before stimulus onset to 20 after stimulus offset (to ignore opto-related artifact bleeding through the red channel) and random noise (from normal distribution centered at 0) added to interpolated timepoints. $F_{tdtomato}$ was then smoothed (moving average with a window of 50 s), and ICA was used (rica function implemented in MATLAB) to extract background and signal components from $F_{tdtomato}$ and $F_{GCAMP}$. Independent component highly correlated to $F_{tdtomato}$ (absolute correlation coefficient >0.9) was considered the background component ($ICA_{Background}(F_{tdtomato}, F_{GCAMP})$), whereas the other component considered the signal component ($ICA_{Signal}(F_{tdtomato}, F_{GCAMP})$). Sign of $ICA_{Signal}(F_{tdtomato}, F_{GCAMP})$ was corrected using the sign of the correlation between $ICA_{Signal}(F_{tdtomato}, F_{GCAMP})$ and $F_{GCAMP}$. For ROIs extracted from pan-neuronal data, we report calcium signals as $ICA_{Signal}(F_{tdtomato}, F_{GCAMP})$ as shown in *Figure 5B*.

We defined responsive ROIs as ROIs with a mean activity during optogenetic stimulation (Ft1) higher than $3\sigma_o$ ($\sigma_o$ - standard deviation of activity during baseline). We then split ROIs into transiently and persistently active units using the mean activity after optogenetic stimulation (Ft2, from stimulus offset to 5 min after stimulus offset), transient ROIs had Ft2 $\leq 3\sigma_o$, while persistent ROIs had Ft2 $> 3\sigma_o$. To evaluate the diversity of these coarse activity types, we hierarchically clustered transient and persistent responses (we evaluated the number of clusters these response types split into using the consensus across Calinski-Harabasz, Silhouette, Gap, and Davies-Bouldin criteria), obtaining two clusters of transient responses and two clusters of persistent responses (*Figure 5C–D*).

For recordings of Dsx+ cell types, we imaged pC1, pC2, pCd1, and pCd2 cells (1–2 groups at a time), located in the dorsal side of the central brain, at a speed of 0.5–0.25 Hz ($0.5 \times 0.5 \times 1$ µm$^3$ - $2.5 \times 2.5 \times 1$ µm$^3$ voxel size). Similarly, for recordings of Fru+ cell types we imaged Fru+ cells (both within the same field of view), located in the dorsal side of the central brain, at a speed of 0.5 Hz ($1.2 \times 1.2 \times 2$ µm$^3$ voxel size). Volumetric time-series of GCaMP6s signal was motion-corrected in the XYZ axes using the NoRMCorre algorithm (*Pnevmatikakis and Giovannucci, 2017*), and temporally resampled to correct for different slice timing across planes of the same volume, and to align timestamps of volumes relative to the start of the optogenetic stimulation (linear interpolation). Dsx + and Fru+ somas were manually segmented by finding the center and edge of each cell body stack by stack (*Deutsch et al., 2019*).

## In vivo whole-cell patch clamp recordings

Stimulus design, recording techniques, and analysis methods are described in detail in *Clemens et al., 2015*. Briefly, virgin female flies (1–2 days old) were mounted and dissected as described previously, and the cell bodies were accessed from the posterior surface of the head and visualized by expressing GFP in pC1-A neurons (see Fly genotype table) – 1 cell was recorded per fly. Raw data were further analyzed using MATLAB (Mathworks). To construct the IPI, sine frequency, and intensity tuning curves, we calculated the integral voltage in response to each stimulus, after

subtracting the baseline voltage. Each response was then normalized by the duration of the stimulus. To compare across individuals, tuning curves were normalized to peak at 1.0.

## Acknowledgements

We thank Barry Dickson, David Anderson, Annegret Falkner and Christa Baker for comments on the manuscript and the entire Murthy lab for helpful discussions. We thank Barry Dickson for sharing the pC1-A split GAL4 line ahead of publication. We thank Stephan Thiberge for assistance with two-photon imaging, Nat Tabris for assistance with software development, Josh Shaevitz for development of modifications to LEAP, Shruthi Ravindranath for assistance with identifying neurons in FlyWire, and Junyu Li for assistance with proofreading behavioral data. We thank Joseph Hsu (Janelia) and other members of the Cambridge *Drosophila* Connectomics Group for contributing to FlyWire tracing of neurons in this study, and Maria Dreher (Janelia) for assistance with matching cell types across datasets. We thank Katie Schretter and Gerry Rubin for exchanging information prior to publication. This study was supported by an NIH BRAIN Initiative RF1 MH117815-01 to MM and HSS and an NIH BRAIN R01 NS104899 and HHMI Faculty Scholar award to MM.

## Additional information

### Funding

| Funder | Grant reference number | Author |
| --- | --- | --- |
| National Institutes of Health | RF1 MH117815-01 | Mala Murthy |
| National Institutes of Health | R01 NS104899 | Mala Murthy |
| Howard Hughes Medical Institute | Faculty Scholar | Mala Murthy |

The funders had no role in study design, data collection and interpretation, or the decision to submit the work for publication.

### Author contributions

David Deutsch, Conceptualization, Data curation, Software, Formal analysis, Investigation, Visualization, Methodology, Writing - original draft, Writing - review and editing; Diego Pacheco, Data curation, Formal analysis, Investigation, Methodology, Writing - original draft; Lucas Encarnacion-Rivera, Ramie Fathy, Elise Ireland, Data curation, Formal analysis; Talmo Pereira, Software, Writing - original draft; Jan Clemens, Formal analysis, Writing - original draft; Cyrille Girardin, Data curation, Investigation, Methodology; Adam Calhoun, Sven Dorkenwald, Claire McKellar, Thomas Macrina, Ran Lu, Kisuk Lee, Nico Kemnitz, Dodam Ih, Manuel Castro, Akhilesh Halageri, Chris Jordan, William Silversmith, Jingpeng Wu, Software; Austin Burke, Data curation; H Sebastian Seung, Software, Funding acquisition, Project administration; Mala Murthy, Conceptualization, Resources, Supervision, Funding acquisition, Writing - original draft, Project administration, Writing - review and editing

### Author ORCIDs

David Deutsch ⓘ https://orcid.org/0000-0002-8587-2435
Talmo Pereira ⓘ https://orcid.org/0000-0001-9075-8365
Jan Clemens ⓘ http://orcid.org/0000-0003-4200-8097
Claire McKellar ⓘ https://orcid.org/0000-0002-3580-7336
Mala Murthy ⓘ https://orcid.org/0000-0003-3063-3389

### Decision letter and Author response

Decision letter https://doi.org/10.7554/eLife.59502.sa1
Author response https://doi.org/10.7554/eLife.59502.sa2

## Additional files

### Supplementary files

• Supplementary file 1. FlyWire cells identified and proofread for this study. This Table contains Fly-Wire segment IDs for each cell, cell names where identified, and links to the cells in FlyWire.

• Transparent reporting form

### Data availability

All data generated or analysed during this study are included in the manuscript and supporting files.

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
