## [Decision Letter]

**Acceptance summary:**

This timely study employs an arsenal of tools, from precise behavioral measurement, to careful anatomical tracing, with optogenetic perturbations and large-scale imaging to identify a subset of neurons in the female *Drosophila* brain that modulate internal state. It links minutes-long persistent changes in behavior with recurrent circuit architecture and persistent neural activity. The paper is a technical tour de force that opens new avenues to explore how lasting behavioral states are instantiated and how these might relate to sustained brain states.

**Decision letter after peer review:**

Thank you for submitting your article "The Neural Basis for a Persistent Internal State in *Drosophila* Females" for consideration by *eLife*. Your article has been reviewed by three peer reviewers, and the evaluation has been overseen by a Reviewing Editor and Laura Colgin as the Senior Editor. The following individuals involved in review of your submission have agreed to reveal their identity: Michael M Yartsev (Reviewer #1); Marta A Moita (Reviewer #3).

The reviewers have discussed the reviews with one another and the Reviewing Editor has drafted this decision to help you prepare a revised submission.

The full reviews are appended below, and we look forward to receiving a revised manuscript and response to reviewers that addresses the points raised by the reviewers. The most serious concerns, raised by multiple revisions, are synthesized and highlighted under "Essential Revisions," below.

As the editors have judged that your manuscript is of interest, but as described below that additional substantive revisions are required before it is published, we would like to draw your attention to changes in our revision policy that we have made in response to COVID-19 (https://elifesciences.org/articles/57162). First, because many researchers have temporarily lost access to the labs, we will give authors as much time as they need to submit revised manuscripts. We are also offering, if you choose, to post the manuscript to bioRxiv (if it is not already there) along with this decision letter and a formal designation that the manuscript is "in revision at *eLife*". Please let us know if you would like to pursue this option. (If your work is more suitable for medRxiv, you will need to post the preprint yourself, as the mechanisms for us to do so are still in development.)

In addition, we are aware of a related bioRxiv preprint by Schretter et al. and we encourage you to refer to that paper in your revision and/or rebuttal where it may be appropriate to strengthen support for the claims.

Essential revisions:

1) The most serious concerns related to the accuracy of the cell-type identification. This concern comprises the bulk of the feedback from reviewer 2, and related points were raised by reviewer 3 (Methodological points 1, 3).

2) Physiological relevance of optogenetically-induced behaviors, particularly in light of the extremely long duration stimulation (5min). This issue was raised by both reviewer 2 (last point) and reviewer 3 (Conceptual point 2). The long stimulation protocol should be directly addressed in the manuscript. Ideally, shorter stimulation times would be included, but at an absolute minimum the limitations need to be recognized. The Anderson lab showed that a 7.5 s stimulation of P1 neurons (the male analogues of pC1) could produce a persistent activity in a downstream population. If Deutsch et al. tried shorter times and they didn't work, this should be discussed. There were related issues about how naturalistic the elicited behaviors were. This point was raised by both reviewer 1 (point 4) and reviewer 3 (Conceptual point 1).

3) Need for additional quantification and statistics was a general concern, highlighted specifically by reviewers 1 and 3 (see reviewer 1, points 2 and 5; reviewer 3 Methodological points 2 and 4).

The remainder of the concerns, such as justification or toning down of 'emotional state' (reviewer 3 Conceptual point 3), precision of driver lines, and the completeness of the EM reconstruction of upstream and downstream connectivity, can likely be dealt with through textual revisions adjusting the strength/justification of the claims.

*Reviewer #1:*

The manuscript by Deutsch and colleagues present an intriguing account for the potential role of persistent neural activity in the female *Drosophila* during social interaction. As a bat researcher I am very jealous by the impressive arsenal employed by the group of researchers ranging from precise behavioral measurement, to carefully anatomical delineation, optical stimulation and through large-scale imaging approaches. Using these, the authors identify an important subset of neurons in the female *Drosophila* brain that are participating in long-term (minutes) changes in female behavior. This work will serve as a cornerstone for continued study of social behavior and communication and beyond. I congratulate the authors on this spectacular body of work.

Substantive comments: My comments primarily aim towards suggestions for clarifications.

1) What behavioral changes, if any, could be driven by the male song played back to the female but without the male actually being present?

2) The authors state that "the weights of 8 out of 17 parameters were significantly different from zero". To ease the reader, it might be helpful for the author to elaborate how significance was corrected for multiple comparisons and false-discovery rate (FDR)?

3) The authors describe an intriguing clustering of distinct behaviors emerging from stimulation. I wonder if the authors can extend their analysis into behavioral sequences? It particular, if such sequences can be described it would be intriguing to know the relationships between their prevalence and delay of stimulation.

4) Further, it would be interesting to know how the behavioral sequences following stimulation compare to those exhibited around copulation independent of stimulation? Basically how "natural are those sequences"? For example, the author could do something like a reverse analysis and set time 'zero' as the time of copulation (or any other major behavioral event to be consider the 'end-point') and then work backwards looking at the sequences of behavioral events that led to the 'end-point'. How similar are the behaviors leading to the 'end-point' behavior under the two conditions? Are they similar and if so, for how long? Perhaps the correlation between the sequence following stimulation and that not following stimulation decays (or increases) as a function of time? Given the wealth of knowledge from this group about the social behavior of flies I think the readers would appreciate seeing a comparison to the behavior following stimulation.

5) In the imaging experiments (Figure 6) the authors note that they clustered ROIs based on response patterns (Figure 6D). How were those clustered? Was it done by eye? If so, perhaps it would be possible to either validate this clustering using a machine-learning or statistical classification approach or alternatively demonstrate the robustness/reproducibility of the responses within a given cluster? Similarly, for the ROI's exhibiting sustained activity on the minutes scale. It would be helpful to get a sense of this variability in a quantitative manner.

6) Did the authors play male song (and other sounds) to the female during the head-fixed imaging experiments? I realize this is a highly unnatural state for the female but it would still be interesting to know if/how auditory stimulation modulated the responses.

*Reviewer #2:*

There is a great deal to like about this study, but I do have a significant concern about the classification of pC1 neurons that I feel must be clarified before making any firm recommendation. In addition I think that setting a lower limit on the duration of optogenetic stimulation required to induce persistent activity would be helpful.

The authors claim there are 5 pC1 cells / brain hemisphere and 7 types in all. Does this mean that some cells are so different from left to right that they form different cell types? Although there are reports of left-right differences in the fly literature, this situation seems unusual. Another explanation is that some cells have been incompletely or incorrectly reconstructed from EM data. The amazing recent advances in *Drosophila* EM connectomics, including the impressive flywire.ai resource presented in this study, means that we are in a somewhat unusual situation in which the exact same pC1 neurons central to this paper have been independently reconstructed from the same brain by two groups (this study; Wang F. et al., 2020, senior author Barry Dickson). Although Deutsch et al. note:

“Because our results are based on automated segmentation followed by manual

proofreading (in 3D), they may differ from results of manual tracing in 2D, even though we have used the same underlying EM dataset.”

If there are large differences in the morphology of reconstructed neurons, then presumably someone is right or wrong. After reviewing the morphologies in these two manuscripts, we believe that the neuron called pC1-Alpha-s in this manuscript is the contralateral homologue of the neuron called pC1e in this manuscript. This neuron does not have a ventromedial branch as presented in Figure 3; but a ventromedial branch is present in the manually traced version of what appears to be exactly the same neuron in Wang F. et al., 2020 – Figure 3D. Based on this comparison we conclude that:

pC1-Alpha-l = pC1d; pC1-Alpha-s = pC1e (Wang F. et al., 2020 nomenclature listed second)

pC1-Med = pC1c; one cell type according to Wang F. et al. – this does not appear to us to affect any of the results in this manuscript.

The presence or absence of this ventromedial branch is significant because the authors use it as a diagnostic feature when determining which EM cell types are present in their genetic driver lines. While describing their driver lines, the authors note:

“pC1-A has a medial projection (red arrow), similar to pC1-Alpha-l neurons found in EM (C); the medial projection was found in 7/7 imaged pC1-A female brains, in both hemispheres. This projection was not found in pC1-S imaged female brains (8/8 brains) – neurons labeled in pC1-S resemble pC1-e neurons found in EM (C).”

If we accept that pC1e and pC1-Alpha-s are indeed the same cell type, then "pC1-A line" most likely contains pC1d and pC1e (since it contains 2 cells per hemisphere). This is also consistent with multi color Flp out data present in Schretter et al., 2020, Supplementary Figure 5E. Furthermore in contrast to the conclusion of Deutsch et al., since it lacks the ventromedial branch, the "pC1-S line" most likely contains something other than pC1e (in Figure 3D it looks like there are both strong and weakly expressing cells and that it might contain all 3 of the remaining pC1 neurons).

This also implies that the behavioral phenotype observed with pC1-A line could be due to either pC1d or pC1e and therefore there is no reason to focus connectivity analyses only on pC1d and not pC1e. And the behavioral experiments done with the pC1-S line might likely reflect the functions of other pC1 types (a, b, c), but not pC1e.

To resolve these issues, further tracing of the neuron currently called pC1e is necessary ideally by comparing it with the published skeleton of pC1e from Wang F. et al., 2020, and/or the Janelia "hemibrain" dataset which now seems to have pC1a-e annotated. Without this, the data presented in Figure 4 is very hard to interpret. An MCFO experiment with both pC1-A and pC1-S could clarify what subtypes (and other neurons in the case of pC1-S) they label, or cite the relevant data for pC1-A from Schretter et al., 2020. In addition the synaptic tracing should probably include both pC1d and pC1e.

As a final note on the subject of EM data, when searching the hemibrain data (neuprint.janelia.org) for pC1 neurons in the course of this review, we noticed that the pC1d neuron in that dataset (which is presumably missing part of its contralateral branch) has 4369 upstream connections and 8945 downstream connections. Furthermore the preprint accompanying that dataset said that they estimate that on average only about a quarter of upstream connections are identified so the true number might be 15-20,000 upstream connections for a single pC1d neurons. In several places Deutsch et al. refer to about 400 pre and postsynaptic sites for their traced pC1-Alpha e.g.:

“After proofreading the pC1-Alpha cell and its input and output cells, and excluding weak connections (using 3 synapses as a threshold, see Materials and methods), we counted 417 presynaptic and 421 postsynaptic sites (Figure 5B, Video 4, and Table 2).”

I think these numbers must refer to presynaptic or postsynaptic partner neurons rather than synaptic sites as otherwise the difference in numbers is too huge. It would be very helpful if the authors could clarify their terminology and/or these differences in synapse numbers. It seems that everyone in the *Drosophila* circuits field will soon need to know how to critically assess EM data.

To help future readers, we would also suggest that the authors use the convention of presenting brain/neuron images in consistent anterior (frontal) rather than posterior views. Furthermore since Wang F. et al., 2020, have already reported pC1 cell types using a complete analysis of the same EM data with accompanying LM work, we would recommend against introducing a different nomenclature.

Another aspect of the paper that raised questions was the length of the stimulation required to evoke persistent neural activity. Do the authors have data about the effects of shorter stimulation? Was there a specific rationale for choosing 5 minutes? In a very similar circuit in male flies, Jung et al., 2019, use 7.5s of P1 stimulation to create minutes long sustained activity in pCd. It is unclear whether 5 minutes of constant excitation can happen in vivo as opposed to experimental manipulations like optogenetic activation.

*Reviewer #3:*

This study addresses the important, poorly understood and poorly defined topic of animals' internal states. This is a timely study that constitutes a technical tour the force that opens new avenues to explore how lasting behavioral states are instantiated and how these might relate to sustained brain states. This study, however, falls short of demonstrating the relationship between the artificially induced brain and behavioral states with natural, endogenous ones, as well as establishing a causal link between the recurrent connectivity, the persistent activity and the behavioral states. I have a few concerns, both at the conceptual and methodological levels.

Conceptual concerns:

Although the authors show that artificial stimulation of a specific set of neurons impacts female receptivity, aggressive behaviors, and neuronal activity in a lasting manner, caution in the interpretation of the reported findings is warranted. I believe a discussion that more openly addresses the short comings of their study is important.

1) The authors devote a section of the Discussion to the finding that activating pC1-int leads to an increase in receptivity while at the same time it triggers aggressive behaviors. They mention in this regard that within pC1 neuronal type there appears to be segregation of neurons that drive courtship behaviors and aggression. Still, alternative explanations, that question the induction of a receptivity state are possible, even if flies end up mating more. Stimulation of pC1-int neurons induces behaviors that normally do not occur in a receptive female. It could be, for example, that a stimulated female is not more receptive, but by displaying aggressive behaviors towards the male, the later becomes aroused and more efficient at mating. The authors should show how activation of the female affects courtship behaviors of the male, including but not exclusively regarding song.

2) The authors use a single form of neuronal stimulation: pulsed light at 100Hz for 5 minutes. It is unclear what kind of neuronal activity it induces during stimulation and how this neuronal activity compares to endogenous activity states in general, and during social interactions in particular. This is especially true in the light of a previous study (Zhou et al., 2014) showing transient activity of pC1 neurons to male song and pheromone (this may be different in a female interacting with a male during courtship). It would have been ideal to at least try different activation patterns, namely shorter stimulation protocols. It may be difficult for the authors to add further experiments with different activation protocols. Therefore, the authors should address this in the Discussion.

3) The authors mention in the Discussion that their observations may be in line with an 'emotional state', as they find lasting states that they claim to be scalable, as they report different decay functions of persistent activity across different brain regions. Although the authors do induce a persistent activity state, evidence for scalability is at best weak. Furthermore, there are multiple features of emotional states, such as somatic responses to the external triggers, that are not addressed in this study. Given that the only robust feature they find is lasting neuronal activity and behaviors, I believe the authors should avoid such claims.

Methodological concerns:

1) For the TNT experiments, Figure 1, the authors use the same control for TNT expression in pC1-int and pCd1 neurons. However, according to the table of genotypes used in this study, it seems that TNT is inserted in different chromosomes for the two experimental lines (2nd chromosome for pC1-int line and 3rd chromosome for pCd1 line). Importantly the control has the TNT insertion in the 3rd chromosome and is thus different from the main line of this study, the pC1-int line. It is also not clear to me if the control corresponds to an empty lex-A line or a parental control. The authors should clarify the controls used and if indeed the control does not have the same insertion sites as the mains experimental line. In this case the experiment should be repeated with the appropriate control, as it is our experience in the lab that these issues are often determinant in the experiment's outcome.

2) Figure 4F shows the probability density distributions of fraction of time spent shoving or chasing, for different experimental lines and different times points. They conclude from these plots that differences in behavior across experimental lines depend on the time point looked at. This is potentially an interesting finding, but I could not find the statistical comparisons that sustain such claim.

3) The authors claim that the neuronal subtype aIP-g-b is the most interconnected for cells that are also reciprocally connected with pC1-alpha. However, Figure 5H seems to show that cells of the aIP-g-c subtype show a similar pattern. It is unclear why the authors are singling out aIP-g-b neurons and how would it be relevant to the claims in the manuscript.

4) Figure 6G shows the activity levels in different brain regions at the 3 tested time points (0,3 and 6 minutes after stimulation). They use this information to say that different brain regions show different decay functions. Again, I could not find the statistical analysis that would be required to make such a claim. This is important as it is one of the pieces of evidence used to suggest that pC1 activation may lead to an emotion-like internal state. It is also not clear to me how different decay functions in different brain regions reflect scalability. To my knowledge scalability typically reflects effects of intensity on output, such as the well-established case of fear studies in rodents where stronger shocks leads to more freezing, or higher levels of corticosterone among other scalable outputs (whether these states correspond to fear is still a matter of debate).

---

## [Author Response]

Essential revisions:1) The most serious concerns related to the accuracy of the cell-type identification. This concern comprises the bulk of the feedback from reviewer 2, and related points were raised by reviewer 3 (Methodological points 1, 3).

We agree with the reviewers and have made the following modifications:

1) We revised the pC1 nomenclature to follow the one used by Wang F. et al., 2020, as suggested by reviewer #2. By performing additional analyses in FlyWire, we have now found a pair of cells (1 per hemisphere) for each one of the pC1 subtypes (pC1a-e) and revised the figures and text accordingly – see revised Figure 3.

2) We now state in the manuscript that the line pC1-A includes both pC1d and pC1e cells, citing Schretter et al., 2020 – this is the same as their line pC1dSS3.

3) Since the pC1-A line includes *both* pC1d and pC1e, we have now added connectome information for pC1e (see new Figure 6—figure supplement 1).

4) By communicating with Katie Schretter and Gerry Rubin, we have now compared the classification of aIPg cells in our work (in FlyWire) with their work (in the hemibrain). We make clear now in the manuscript the correspondence between aIPg cell types in the two studies.

5) We used the hemibrain nomenclature for naming cells we found in FlyWire, wherever a clear match was found based on morphology , to allow easier comparisons between the connectivity diagrams of the two manuscripts (see new Figure 6 and Figure 6—figure supplement 1).

2) Physiological relevance of optogenetically-induced behaviors, particularly in light of the extremely long duration stimulation (5min). This issue was raised by both reviewer 2 (last point) and reviewer 3 (Conceptual point 2). The long stimulation protocol should be directly addressed in the manuscript. Ideally, shorter stimulation times would be included, but at an absolute minimum the limitations need to be recognized. The Anderson lab showed that a 7.5 s stimulation of P1 neurons (the male analogues of pC1) could produce a persistent activity in a downstream population. If Deutsch et al. tried shorter times and they didn't work, this should be discussed. There were related issues about how naturalistic the elicited behaviors were. This point was raised by both reviewer 1 (point 4) and reviewer 3 (Conceptual point 1).

We have now collected new data to address this concern. We activated pC1d/e cells (using the pC1-A split GAL4 line) for shorter durations of 2 minutes or 30 seconds in solitary females, and measured the effect on female behavior following introduction of a male, as before. Females showed a significant level of persistent shoving and chasing (aggressive and male-like behaviors) following 2 minute activation (Figure 4J), but only very few flies showed persistent shoving or chasing following 30 second activation. We have also included data on 5 minute activation of other Dsx+ cell types, such as pCd neurons and pC1 neurons other than pC1d and pC1e (labeled in the pC1-S split GAL4 line), and show that we do not see persistent effects on female behavior with a male (Figure 2—figure supplement 2A) or significant persistent brain activity (Figure 6E; Figure 6—figure supplement 1) following activation of these neurons – this suggests that the persistent changes in behavior we observe are not a consequence of 5 minute activation of any Dsx+ cell type, but are specific to pC1d/e. We also have added new data from pan-neuronal calcium imaging following the shorter 2 minute activation of pC1d/e neurons. We observed clear persistent activity (see new Figure 7), although its distribution was sparser compared to 5-minute activation. We also recorded GCaMP responses in Dsx+ neurons and Fru+ neurons (new Figure 8). Persistent responses were found only in specific subpopulations (e.g., in Dsx+ pC1 neurons, or Fru+ putative aIPg neurons), demonstrating that persistent responses are found only in specific cell types. Finally, we performed patch clamp recordings from pC1d/e neurons in virgin females and found auditory responses to courtship song elements (new Figure 4—figure supplement 1). This suggests that pC1d/e neurons would be activated during courtship as females listen to male song – males sing 100s of song bouts during a typical courtship session (see Coen et al., 2014), but whether this activation can induce a persistent state on its own or in combination with other cues, remains to be determined.

3) Need for additional quantification and statistics was a general concern, highlighted specifically by reviewers 1 and 3 (see reviewer 1, points 2 and 5; reviewer 3 Methodological points 2 and 4).

We have made the modifications requested by the reviewers (see specific responses below).

The remainder of the concerns, such as justification or toning down of 'emotional state' (reviewer 3 Conceptual point 3), precision of driver lines, and the completeness of the EM reconstruction of upstream and downstream connectivity, can likely be dealt with through textual revisions adjusting the strength/justification of the claims.

We have made the revisions requested by the reviewers (see specific responses below).

Reviewer #1:The manuscript by Deutsch and colleagues present an intriguing account for the potential role of persistent neural activity in the female *Drosophila* during social interaction. As a bat researcher I am very jealous by the impressive arsenal employed by the group of researchers ranging from precise behavioral measurement, to carefully anatomical delineation, optical stimulation and through large-scale imaging approaches. Using these, the authors identify an important subset of neurons in the female *Drosophila* brain that are participating in long-term (minutes) changes in female behavior. This work will serve as a cornerstone for continued study of social behavior and communication and beyond. I congratulate the authors on this spectacular body of work.Substantive comments: My comments primarily aim towards suggestions for clarifications.1) What behavioral changes, if any, could be driven by the male song played back to the female but without the male actually being present?

We thank the reviewer for these comments. This is an interesting question about the specific sensory stimuli that affect female behavior *following* pC1 activation, or once the persistent internal state is induced. While we have not yet dissected which sensory responses change and how, we did examine the persistent effect of pC1-Int activation on female responses to pulse song in a playback assay. Preliminarily, we found that female responses to song were different during the 5min optogenetic activation of pC1 neurons versus the 5min in between stimuli – however, we would need to perform additional experiments to interpret these results and compare them to our results during natural behavior, and so we have not included them in the revision. We should point out though that the companion study from Schretter et al. show that aIPg neurons, the neurons recurrently connected with pC1d/e, receive a large number of visual inputs and that aggressive behaviors driven by aIPg-b neurons are different in light versus dark. This suggests that the internal state changes driven by pC1d/e affect visual responses via aIPg neurons, and we now state this explicitly in the Discussion: “‘female approaching’ (Figure 2—figure supplement 3A), begins from a distance greater than 4 body lengths from the male fly (a distance at which it may be difficult to discern male from female [Borst, 2009]) and often ends with shoving or circling (see Video 3), we hypothesize that pC1 activation most likely drives persistent behaviors towards another fly, and not specifically a male or female fly, consistent with (Schretter et al., 2020).”

2) The authors state that "the weights of 8 out of 17 parameters were significantly different from zero". To ease the reader, it might be helpful for the author to elaborate how significance was corrected for multiple comparisons and false-discovery rate (FDR)?

As there are 17 comparisons (Figure 2B and Figure 2—figure supplement 1A), we used Bonferroni corrections for multiple comparisons in the original version, but neglected to mention this in the figure legends – fixed now.

3) The authors describe an intriguing clustering of distinct behaviors emerging from stimulation. I wonder if the authors can extend their analysis into behavioral sequences? It particular, if such sequences can be described it would be intriguing to know the relationships between their prevalence and delay of stimulation.

We thank the reviewer for the suggestion – we have now quantified the transitions between shoving, chasing, and other (not shoving and not chasing) behavioral states (using the data analyzed in JAABA). This is now plotted in Figure 2—figure supplement 2C. This analysis reveals the probability of transitioning between states, and how this changes with different delays following pC1 activation. As evident from the new figure, the transition probability from shoving to chasing more than doubles in the d3 condition compared to the d0 and d6 conditions. The result is summarized in the manuscript: “In many cases females transitioned directly between shoving and chasing, and the conditioned probability (transition probability, given that a transition occurred) for shoving chasing was more than double at the d3 condition compared to the d0 and d6 conditions.”.

4) Further, it would be interesting to know how the behavioral sequences following stimulation compare to those exhibited around copulation independent of stimulation? Basically how "natural are those sequences"? For example, the author could do something like a reverse analysis and set time 'zero' as the time of copulation (or any other major behavioral event to be consider the 'end-point') and then work backwards looking at the sequences of behavioral events that led to the 'end-point'. How similar are the behaviors leading to the 'end-point' behavior under the two conditions? Are they similar and if so, for how long? Perhaps the correlation between the sequence following stimulation and that not following stimulation decays (or increases) as a function of time? Given the wealth of knowledge from this group about the social behavior of flies I think the readers would appreciate seeing a comparison to the behavior following stimulation.

Virgin female shoving or chasing behaviors directed towards a courting male are rare (see Figure 2F, first panel). Mated females are known to show aggressive behaviors such as shoving/fencing (see e.g., Cook and Connolly, 1973), but we did not examine mated females in our experiments. Females also show these aggressive behaviors during female-female fights over food sources (e.g., Nilsen et al., 2004). We have now added text in the Discussion to make this point clearer: “In *Drosophila*, female-female fights over food source are strongly stimulated by the receipt of sperm at mating (Bath et al., 2017), and include both patterns that are common with male aggression (such as shoving and fencing) and female-only patterns (Nilsen et al., 2004). Female-male aggression was reported in the context of rejecting behavior in mated, immature, or older females (Cook and Connolly, 1973).”

It will be interesting to examine, as a future direction, how the arousal state we uncover here is triggered under more natural conditions. We found that the highest levels of pC1 activation enhance receptivity but have an opposing effect on responses to male song (speeding females up instead of slowing them down). Slightly lower levels of pC1 activation (following a delay) bias females toward aggression and male-like behaviors. These effects on behavior may not naturally co-occur, but our optogenetic activation paradigm uncovers the scaleable relationship between activation of different pC1 subtypes, their individual levels of activity, and distinct behavioral programs.

5) In the imaging experiments (Figure 6) the authors note that they clustered ROIs based on response patterns (Figure 6D). How were those clustered? Was it done by eye? If so, perhaps it would be possible to either validate this clustering using a machine-learning or statistical classification approach or alternatively demonstrate the robustness/reproducibility of the responses within a given cluster? Similarly, for the ROI's exhibiting sustained activity on the minutes scale. It would be helpful to get a sense of this variability in a quantitative manner.

To determine the minimum number of clusters comprising either transient or persistent ROIs, we used hierarchical clustering (as explained in the Materials and methods section ‘in vivo whole-brain calcium imaging’). The number of clusters was determined using the consensus across Calinski-Harabasz, Silhouette, Gap, and Davies-Bouldin criteria. We have now added a description in the Materials and methods: “the GCaMP6s signal was 3D-ROI segmented using the Constrained Nonnegative Matrix Factorization (CNMF) algorithm to obtain temporal traces and spatial footprints per segmented ROI as implemented in CaImAn (Giovannucci et al., 2019; Pacheco et al., 2019). […] Code to perform these processing steps are available at https://github.com/murthylab/FlyCaImAn”

6) Did the authors play male song (and other sounds) to the female during the head-fixed imaging experiments? I realize this is a highly unnatural state for the female but it would still be interesting to know if/how auditory stimulation modulated the responses.

While we did not examine how pC1 activation changes neural responses to courtship song, a recent study did examine this (Wang K. et al., 2020) – they found that pC1a subtype activation enhances auditory responses of Dsx+ descending neurons that drive vaginal plate opening, such that listening to the male pulse song, if pC1a neurons are active, causes her to ‘become receptive’ by opening the vaginal plate. In our study, we focused on pC1d/e neurons, and we do not yet know how activity of these neurons might alter auditory activity. However, Schretter et al., 2020, have examined visual inputs to aIPg neurons and finds they are numerous. This suggests that pC1d/e activation affects visuomotor transformations via aIPg neurons, and we now mention this in the Discussion. In the behavior, we suspect that it is visual cues from the male that drive her aggressive and male-like chasing behaviors. pC1a neurons may be the subtype that cause enhanced receptivity, but it is curious that we find this co-occurs with speeding up to male song (a rejection behavior) versus slowing down. Future work will have to examine the complex interplay between male acoustic cues, male visual cues, and female responses to the male as her internal state changes.

Reviewer #2:There is a great deal to like about this study, but I do have a significant concern about the classification of pC1 neurons that I feel must be clarified before making any firm recommendation. In addition I think that setting a lower limit on the duration of optogenetic stimulation required to induce persistent activity would be helpful.The authors claim there are 5 pC1 cells / brain hemisphere and 7 types in all. Does this mean that some cells are so different from left to right that they form different cell types? Although there are reports of left-right differences in the fly literature, this situation seems unusual. Another explanation is that some cells have been incompletely or incorrectly reconstructed from EM data. The amazing recent advances in *Drosophila* EM connectomics, including the impressive flywire.ai resource presented in this study, means that we are in a somewhat unusual situation in which the exact same pC1 neurons central to this paper have been independently reconstructed from the same brain by two groups (this study; Wang F. et al., 2020, senior author Barry Dickson). Although Deutsch et al. note:“Because our results are based on automated segmentation followed by manualproofreading (in 3D), they may differ from results of manual tracing in 2D, even though we have used the same underlying EM dataset.”If there are large differences in the morphology of reconstructed neurons, then presumably someone is right or wrong. After reviewing the morphologies in these two manuscripts, we believe that the neuron called pC1-Alpha-s in this manuscript is the contralateral homologue of the neuron called pC1e in this manuscript. This neuron does not have a ventromedial branch as presented in Figure 3; but a ventromedial branch is present in the manually traced version of what appears to be exactly the same neuron in Wang F. et al., 2020 – Figure 3D. Based on this comparison we conclude that:pC1-Alpha-l = pC1d; pC1-Alpha-s = pC1e (Wang F. et al., 2020 nomenclature listed second)pC1-Med = pC1c; one cell type according to Wang F. et al. – this does not appear to us to affect any of the results in this manuscript.The presence or absence of this ventromedial branch is significant because the authors use it as a diagnostic feature when determining which EM cell types are present in their genetic driver lines. While describing their driver lines, the authors note:“pC1-A has a medial projection (red arrow), similar to pC1-Alpha-l neurons found in EM (C); the medial projection was found in 7/7 imaged pC1-A female brains, in both hemispheres. This projection was not found in pC1-S imaged female brains (8/8 brains) – neurons labeled in pC1-S resemble pC1-e neurons found in EM (C).”

From additional proofreading of neurons in FlyWire, we have come to the same conclusion as the reviewer about the pC1d and pC1e cell types. We have identified 2 cells (one in each hemisphere) for the other three pC1 cell types identified in Wang F. et al., 2020 – so for pC1a-e we have 10 cell types in total. We have now updated Figure 3 accordingly and have used the pC1 names from Wang F. et al., 2020 in our study.

We do point out, however, that a small branch is missing in one of the pC1e cells, which made us previously differentiate between pC1e (Wang F. et al., 2020) and what we had called pC1-Alpha-s (we looked carefully for it in the EM dataset), but we agree that both cells look similar otherwise. For pC1c – there is some variability between the cells in the two hemispheres, as we mention in the figure legend, but we agree that these two cells are morphologically similar enough to be classified as the same cell type.

If we accept that pC1e and pC1-Alpha-s are indeed the same cell type, then "pC1-A line" most likely contains pC1d and pC1e (since it contains 2 cells per hemisphere). This is also consistent with multi color Flp out data present in Schretter et al., 2020, Supplementary Figure 5E.

Our pC1-A line is the same as the pC1dSS3 line from Schretter et al., 2020. According to Schretter et al., this line contains one pC1d and one pC1e cell in each hemibrain. We now cite Schretter et al., 2020 in our manuscript when we first describe the split GAL4 line: “The pC1-A line (same as pC1dSS3 (Schretter et al., 2020)) includes two cells per hemisphere, one pC1d and one pC1e cell.”

Furthermore in contrast to the conclusion of Deutsch et al., since it lacks the ventromedial branch, the "pC1-S line" most likely contains something other than pC1e (in Figure 3D it looks like there are both strong and weakly expressing cells and that it might contain all 3 of the remaining pC1 neurons).

We again agree with the reviewer here, and have now made it clear in the main text and in the Figure 3 legend, that the pC1-S split line likely contains a mix of pC1a-c cells. pC1c is less likely as pC1-S doesn’t contain the medial pC1c projection – we couldn’t find it after looking at the N=7 female brains we imaged for this line. We now mention this in the text: “The medial projection that is unique to the pC1c subtype was missing in all the 8 pC1-S flies we imaged, therefore it is likely that pC1-S includes only pC1a and/or pC1b cells.”.

This also implies that the behavioral phenotype observed with pC1-A line could be due to either pC1d or pC1e and therefore there is no reason to focus connectivity analyses only on pC1d and not pC1e. And the behavioral experiments done with the pC1-S line might likely reflect the functions of other pC1 types (a, b, c), but not pC1e.To resolve these issues, further tracing of the neuron currently called pC1e is necessary ideally by comparing it with the published skeleton of pC1e from Wang F. et al., 2020, and/or the Janelia "hemibrain" dataset which now seems to have pC1a-e annotated. Without this, the data presented in Figure 4 is very hard to interpret.The results presented in Schretter et al., 2020, implicate pC1d not pC1e in aggressive phenotypes driven by neural activation – these results are based on split GAL4 lines specific for pC1d or pC1e. We now cite this work on this point in our manuscript “The pC1-A line (same as pC1dSS3 (Schretter et al., 2020)) includes two cells per hemisphere, one pC1d and one pC1e cell. Schretter et al., 2020 have demonstrated that optogenetic activation of genetic lines that contain pC1d, but not pC1e, drive female aggressive behaviors, such as those we refer to here as ‘shoving”. However, since the behavioral experiments presented in our study differ from those in Schretter et al., especially with regard to induction of a persistent state, we acknowledge that a possible contribution of pC1e could not be ruled out (as we now state, see e.g., the section under “Activation of pC1d/e neurons drives persistent neural activity”), and we have therefore added connectivity analysis for pC1e, as suggested by the reviewer (new Figure 6—figure supplement 1).An MCFO experiment with both pC1-A and pC1-S could clarify what subtypes (and other neurons in the case of pC1-S) they label, or cite the relevant data for pC1-A from Schretter et al., 2020. In addition the synaptic tracing should probably include both pC1d and pC1e.

We now cite Schretter et al., 2020, for the fact that pC1-A (their pC1SS3) contains both pC1d and pC1e, “A companion study (Schretter et al., 2020) demonstrates that while pC1-A (their pC1dSS3) labels both pC1d and pC1e, only pC1d drives female aggressive behaviors”. Our confocal imaging shows that pC1-S does not contain the pC1d/e medial branches (Figure 6C-D) – given that these images also do not show the medial branches of pC1c cell types, we conclude that this line most likely labels pC1a-b cell types. We make this clearer in the text (Results).

As a final note on the subject of EM data, when searching the hemibrain data (neuprint.janelia.org) for pC1 neurons in the course of this review, we noticed that the pC1d neuron in that dataset (which is presumably missing part of its contralateral branch) has 4369 upstream connections and 8945 downstream connections. Furthermore the preprint accompanying that dataset said that they estimate that on average only about a quarter of upstream connections are identified so the true number might be 15-20,000 upstream connections for a single pC1d neurons. In several places Deutsch et al. refer to about 400 pre and postsynaptic sites for their traced pC1-Alpha e.g.:“After proofreading the pC1-Alpha cell and its input and output cells, and excluding weak connections (using 3 synapses as a threshold, see Materials and methods), we counted 417 presynaptic and 421 postsynaptic sites (Figure 5B, Video 4, and Table 2).”I think these numbers must refer to presynaptic or postsynaptic partner neurons rather than synaptic sites as otherwise the difference in numbers is too huge. It would be very helpful if the authors could clarify their terminology and/or these differences in synapse numbers. It seems that everyone in the *Drosophila* circuits field will soon need to know how to critically assess EM data.

Our manual identification of synapses for pC1d (formerly pC1-Alpha) in FlyWire revealed 417 and 421 pre/postsynaptic sites, after using a threshold for only including neurons with a minimum number of synapses (3) with pC1d, as described in the manuscript. We also used automated synapse detection using Buhmann et al., 2019 synapses and found 4522 and 10916 presynaptic and postsynaptic synaptic sites to pC1d, numbers not very different from the numbers mentioned in the hemibrain. After using the thresholds described in better detail now in the revised manuscript: “We re-evaluated synaptic partners using automatic detection, and focused on cells with strong connections with pC1d using two criteria: (1) minimum of 6 synapses, (2) the cell belongs to a cell type (based on morphology) with at least one cell with 15 synapses or more with pC1d.”, We found 5255 and 2474 presynaptic and postsynaptic sites for the same pC1d cell (see Table 1). We have now added the list of input and output cells to pC1d and pC1e in Table 1, as well as publicly accessible links to these cells (Table 1, last tab). The method we used for manual detection of synapses was unbiased but not exhaustive. Still, both the automatic and manual methods led to identification of aIPg as the major output cell type of pC1d, with some aIPg cells showing strong recurrent connectivity with pC1d.

To help future readers, we would also suggest that the authors use the convention of presenting brain/neuron images in consistent anterior (frontal) rather than posterior views. Furthermore since Wang F. et al., 2020, have already reported pC1 cell types using a complete analysis of the same EM data with accompanying LM work, we would recommend against introducing a different nomenclature.

After further proofreading, we have now modified the pC1 nomenclature to be consistent with Wang F. et al., 2020. We also made an effort to find morphologically similar cells in the hemibrain for aIPg and for pC1d/pC1e input and output cells. Whenever possible, we use the hemibrain nomenclature. We summarize the comparison in Table 1, in Figures 6 and in Figure 6—figure supplement 1. However, it is important to remember that the naming in the hemibrain is based on both morphology and connectivity, while we sorted cells into types in FlyWire based on morphology alone. We added anterior views in the relevant figures and used the same view for hemibrain and FlyWire where we compare the two. In some cases, we kept the posterior view, as it is consistent with the view used for two-photon imaging, and as our neurons of interest (pC1, aIPg) are all located in the posterior part of the brain.

Another aspect of the paper that raised questions was the length of the stimulation required to evoke persistent neural activity. Do the authors have data about the effects of shorter stimulation? Was there a specific rationale for choosing 5 minutes? In a very similar circuit in male flies, Jung et al., 2019, use 7.5s of P1 stimulation to create minutes long sustained activity in pCd. It is unclear whether 5 minutes of constant excitation can happen in vivo as opposed to experimental manipulations like optogenetic activation.

In Jung et al., 2020, a shorter stimulation (3 pulses, 5s each, with 25ms inter-pulse interval) was used to drive persistent neural activity lasting minutes. While that stimulation was overall shorter in duration, it lasted 1.5 minutes in total. In our case, we used stronger activation (5 minute duration), but inserted up to a 6 minute delay between activation and introduction of a male (to assay behavior). To address the reviewer concerns, we have now added new data – 2 minute and 30 second stimulus duration conditions for the behavioral experiments and found significant shoving and chasing following 2 minutes activation, but very little following 30 second activation (now consistent with Schretter et al. – that study does not observe persistent effects on behavior following 30sec of pC1d activation). We also measured pan-neuronal responses to 2 minute activation of pC1d/e cells and found persistent activity, albeit reduced when compared with 5 min activation. Taken together, this suggests that shorter activation of pC1d/e (2 min) still drives persistent changes in behavior and persistent neural activity – we hypothesize this arises by recruiting aIPg neurons and we mention this in the Discussion.

Reviewer #3:This study addresses the important, poorly understood and poorly defined topic of animals' internal states. This is a timely study that constitutes a technical tour the force that opens new avenues to explore how lasting behavioral states are instantiated and how these might relate to sustained brain states. This study, however, falls short of demonstrating the relationship between the artificially induced brain and behavioral states with natural, endogenous ones, as well as establishing a causal link between the recurrent connectivity, the persistent activity and the behavioral states. I have a few concerns, both at the conceptual and methodological levels.Conceptual concerns:Although the authors show that artificial stimulation of a specific set of neurons impacts female receptivity, aggressive behaviors, and neuronal activity in a lasting manner, caution in the interpretation of the reported findings is warranted. I believe a discussion that more openly addresses the short comings of their study is important.1) The authors devote a section of the Discussion to the finding that activating pC1-int leads to an increase in receptivity while at the same time it triggers aggressive behaviors. They mention in this regard that within pC1 neuronal type there appears to be segregation of neurons that drive courtship behaviors and aggression. Still, alternative explanations, that question the induction of a receptivity state are possible, even if flies end up mating more. Stimulation of pC1-int neurons induces behaviors that normally do not occur in a receptive female. It could be, for example, that a stimulated female is not more receptive, but by displaying aggressive behaviors towards the male, the later becomes aroused and more efficient at mating. The authors should show how activation of the female affects courtship behaviors of the male, including but not exclusively regarding song.

The effect of pC1-Int activation on female receptivity peaks at the d0 condition (following 5 min activation), while the effect on female aggression peaks at the d3 condition (following 5 min activation and a 3min delay), suggesting that the change in mating probability following pC1-Int activation is not simply an indirect effect of changes in male courtship behavior (in response to aggressive females) – we now make this clearer in the text: “These behaviors typically peaked in the d3 condition, where they remained high after male introduction. In contrast, the effect on female receptivity and female responses to male song, were both strongest in the d0 condition, ruling out the possibility that the effect on female receptivity is an indirect consequence of modified male courtship behavior in response to changes in female behaviors.”. Moreover, in Figure 1—figure supplement 1 we show that male song patterns are largely similar in the d0 condition and the control, indicating that in this condition – when there is also a significant effect of pC1-Int activation on female receptivity, male singing (which drives vaginal plate opening in females – see Wang F. et al., 2020b, does not change. While our assay afforded us the opportunity to record song information (using microphones), we did not examine other male behaviors (such as copulation behaviors, which would have required different camera positions). Lastly, activating pC1-A (only pC1d/e neurons) drives female shoving and chasing, without a significant effect on mating probability. We modified the text to make this point clearer: “Activation of either line pC1-A or pC1-S did not affect female receptivity (Figure 4A, D), but activation of pC1-A drove persistent shoving and chasing (Figure 4B-C), while activation of pC1-S did not (Figure 4E).”), and have now modified the text of the Discussion.

2) The authors use a single form of neuronal stimulation: pulsed light at 100Hz for 5 minutes. It is unclear what kind of neuronal activity it induces during stimulation and how this neuronal activity compares to endogenous activity states in general, and during social interactions in particular.

Our experiments involving neural imaging (current Figures 7 and 8) involved imaging *both* during the 5 min of activation and after. We analyze pan-neuronal DF/F levels over the 5min of pC1d/e activation and compare this to a recent study of pan-neuronal responses to auditory stimuli (Pacheco et al., 2019). We find that the DF/F values following activation fall within the distribution of DF/F values in response to sensory stimuli – this suggests that activity levels over 5min activation are not aberrant (new Figure 7—figure supplement 1C). We also have behavioral data following 5min activation of pCd1 neurons and pC1a-b neurons (labeled in line pC1-S) – see Figure 4E and new Figure 2—figure supplement 2A – the results show that the persistent behavioral phenotypes we observe are specific to pC1d/e activation (in lines pC1-Int and pC1-A). Second we included behavioral data on 2min and 30sec activation (of the pC1-A line) and find that persistent effects on behavior are still observed following 2min activation (new Figure 4J). Finally we imaged neural activity following 2min activation of pC1d/e neurons and show that we still observe ROIs with persistent activity (new Figures 7I-J), although this activity is sparser.

This is especially true in the light of a previous study (Zhou et al., 2014) showing transient activity of pC1 neurons to male song and pheromone (this may be different in a female interacting with a male during courtship). It would have been ideal to at least try different activation patterns, namely shorter stimulation protocols. It may be difficult for the authors to add further experiments with different activation protocols. Therefore, the authors should address this in the Discussion.

Please see our response above. To determine how pC1d/e neurons might be naturally triggered, we have now included patch clamp recordings from these neurons in virgin females, and find auditory responses to courtship song features (new Figure 4—figure supplement 1). This suggests that pC1d/e neurons would be activated during courtship as females listen to male song – males sing 100s of song bouts during a typical courtship session (see Coen et al., 2014), but whether this activation can induce a persistent state on its own or in combination with other cues, remains to be determined.

3) The authors mention in the Discussion that their observations may be in line with an 'emotional state', as they find lasting states that they claim to be scalable, as they report different decay functions of persistent activity across different brain regions. Although the authors do induce a persistent activity state, evidence for scalability is at best weak. Furthermore, there are multiple features of emotional states, such as somatic responses to the external triggers, that are not addressed in this study. Given that the only robust feature they find is lasting neuronal activity and behaviors, I believe the authors should avoid such claims.

There are two lines of evidence that link pC1 activity to a scaleable effect on behavior: 1) We found that the highest levels of pC1 activation drive receptivity but have an opposing effect on responses to male song (speeding females up instead of slowing them down). Slightly lower levels of pC1 activation (following a delay) bias females toward aggression and male-like behaviors (see Figures 1F-G and 2D-F) 2) We have now tested a shorter pC1-A activation of 2 minutes for both the behavior and neural imaging, and find persistent, but reduced, effects on behavior (new Figure 4J) and neural activity (new Figure 7J). We now simply state that our results are “similar to” emotion states observed in other systems: “These data suggest that female pC1 neurons can drive an arousal state, similar to male P1 neurons, but whether female pC1 neurons can drive *persistent* changes in behavior and *persistent* neural activity, has not yet been investigated.”).

Methodological concerns:1) For the TNT experiments, Figure 1, the authors use the same control for TNT expression in pC1-int and pCd1 neurons. However, according to the table of genotypes used in this study, it seems that TNT is inserted in different chromosomes for the two experimental lines (2nd chromosome for pC1-int line and 3rd chromosome for pCd1 line). Importantly the control has the TNT insertion in the 3rd chromosome and is thus different from the main line of this study, the pC1-int line. It is also not clear to me if the control corresponds to an empty lex-A line or a parental control. The authors should clarify the controls used and if indeed the control does not have the same insertion sites as the mains experimental line. In this case the experiment should be repeated with the appropriate control, as it is our experience in the lab that these issues are often determinant in the experiment's outcome.

The genotype mentioned for the pC1-Int TNT inactivation in Table 1 was incorrect and we have now fixed this error. The same parent strain was used for making the pC1-Int and pC1d1 – TNT flies, and in both – as in the control – the TNT is on the 3rd chromosome, and the LexAop – on the 2nd.

2) Figure 4F shows the probability density distributions of fraction of time spent shoving or chasing, for different experimental lines and different times points. They conclude from these plots that differences in behavior across experimental lines depend on the time point looked at. This is potentially an interesting finding, but I could not find the statistical comparisons that sustain such claim.

We have now added the statistics and stated explicitly which differences are significantly different and which ones are not in the figure legend. We modified the main text accordingly:

“We found that levels of female chasing are increased with pC1-A activation (relative to pC1-Int) at d3. The levels of shoving are slightly decreased at d0, but this effect was not statistically significant.”

3) The authors claim that the neuronal subtype aIP-g-b is the most interconnected for cells that are also reciprocally connected with pC1-alpha. However, Figure 5H seems to show that cells of the aIP-g-c subtype show a similar pattern. It is unclear why the authors are singling out aIP-g-b neurons and how would it be relevant to the claims in the manuscript.

We assume the reviewer was referring to the sentence “the most interconnected aIP-g-b cells being the ones that are reciprocally connected to pC1-Alpha”. What we meant here was not that aIP-g-b cells are more interconnected than aIP-g-a or aIP-g-c cells, but rather that out of the group of all aIP-g-b cells, the specific aIP-g-b cells that are reciprocally connected to pC1-Alpha-l (now pC1d; these cells are indicated with a red line in Figure 5G) are also interconnected with other aIPg-b cells. We modified the text (“Some cells from the aIPg-b,c groups are also interconnected (Figure 5G). Interestingly, the most interconnected cells within the aIPg-b group were also the ones that are reciprocally connected to pC1d”) to clear up the confusion.

4) Figure 6G shows the activity levels in different brain regions at the 3 tested time points (0,3 and 6 minutes after stimulation). They use this information to say that different brain regions show different decay functions. Again, I could not find the statistical analysis that would be required to make such a claim.

We have now included statistical analysis in new Figure 7—figure supplement 1B.

This is important as it is one of the pieces of evidence used to suggest that pC1 activation may lead to an emotion-like internal state. It is also not clear to me how different decay functions in different brain regions reflect scalability. To my knowledge scalability typically reflects effects of intensity on output, such as the well-established case of fear studies in rodents where stronger shocks leads to more freezing, or higher levels of corticosterone among other scalable outputs (whether these states correspond to fear is still a matter of debate).

We have also now included new data from shorter activations of pC1d/e neurons in line pC1-A – new Figures 4J and 7I-J. We test both effects on behavior (Figure 4J) and on persistent neural activity (Figure 7I-J). In both cases, we observe persistent behavior and neural activity with 2 minute (versus 5 minute) activation, but the amount of behavior or neural activity is reduced. This also supports our claim of scalability. However, we have now adjusted our wording on this point in the Discussion, to tone down our claims: “In general, effects on behavior of such emotion states are thought to scale with levels of persistent neural activity (Hoopfer et al., 2015; Lee et al., 2014). […] These effects on behavior may not naturally co-occur, but our optogenetic activation paradigm uncovers the scalable relationship between activation of different pC1 subtypes, their individual levels of activity, and distinct behavioral programs. ”.